# Task-related hemodynamic responses in human early visual cortex are modulated by task difficulty and behavioral performance

Charlie S Burlingham[1]*, Minyoung Ryoo[2], Zvi N Roth[2], Saghar Mirbagheri[3], David J Heeger[4], Elisha P Merriam[2]

[1]Department of Psychology, New York University, New York, United States; [2]Laboratory of Brain and Cognition, National Institute of Mental Health, National Institutes of Health, Bethesda, United States; [3]Graduate Program in Neuroscience, University of Washington, Seattle, United States; [4]Department of Psychology and Center for Neural Science, New York University, New York, United States

*For correspondence:
charlie.burlingham@nyu.edu

**Competing interest:** The authors declare that no competing interests exist.

**Abstract** Early visual cortex exhibits widespread hemodynamic responses in the absence of visual stimulation, which are entrained to the timing of a task and not predicted by local spiking or local field potential. Such task-related responses (TRRs) covary with reward magnitude and physiological signatures of arousal. It is unknown, however, if TRRs change on a trial-to-trial basis according to behavioral performance and task difficulty. If so, this would suggest that TRRs reflect arousal on a trial-to-trial timescale and covary with critical task and behavioral variables. We measured functional magnetic resonance imaging blood-oxygen-level-dependent (fMRI-BOLD) responses in the early visual cortex of human observers performing an orientation discrimination task consisting of separate easy and hard runs of trials. Stimuli were presented in a small portion of one hemifield, but the fMRI response was measured in the ipsilateral hemisphere, far from the stimulus representation and focus of spatial attention. TRRs scaled in amplitude with task difficulty, behavioral accuracy, reaction time, and lapses across trials. These modulations were not explained by the influence of respiration, cardiac activity, or head movement on the fMRI signal. Similar modulations with task difficulty and behavior were observed in pupil size. These results suggest that TRRs reflect arousal and behavior on the timescale of individual trials.

## Editor's evaluation

This work is of general interest to those using hemodynamic imaging, such as fMRI, to study the brain. A hemodynamic signature that is modulated by arousal level changes on a trial-to-trial basis, such as those evoked by a difficult task, both provides insight into arousal influences on cortical activity and characterize a prominent signal in hemodynamic data that is rarely considered.

## Introduction

Widespread hemodynamic responses, time-locked to trial onsets, occur in the absence of a visual stimulus in early visual cortex, the earliest site of visual cortical processing (*Sirotin and Das, 2009*). These 'task-related' responses (TRRs) have been reported in awake macaques (*Sirotin and Das, 2009*; *Cardoso et al., 2012*; *Cardoso et al., 2019*) using intrinsic signal optical imaging, and in humans using BOLD-fMRI (*Ress et al., 2000*; *Jack et al., 2006*; *Roth et al., 2020*). Unlike stimulus-evoked

responses in V1, they are poorly predicted by changes in mean firing rates or local field potential (LFP) amplitudes (*Sirotin and Das, 2009*), and unlike spatial attentional responses (*Roth et al., 2020*; *Tootell et al., 1998*; *Gandhi et al., 1999*; *Herrmann et al., 2010*), they are spatially diffuse, extending far beyond the focus of spatial attention (*Sirotin and Das, 2009*; *Roth et al., 2020*). Previous studies demonstrate that TRRs are modulated in amplitude by reward and correlate with heart rate and pupil size (*Cardoso et al., 2019*; *Roth et al., 2020*), suggesting that they reflect arousal. These findings raise the question of whether TRRs in early visual cortex change from trial to trial according to task difficulty and behavioral performance, variables linked to arousal level (*Burlingham et al., 2022*).

In this study, we measured TRRs in human early visual cortex while observers performed a visual orientation discrimination task varying in difficulty, building on the task protocol introduced by *Roth et al., 2020*. *Roth et al., 2020* averaged fMRI responses over many trials and observed a positive relationship between reward magnitude and trial-averaged TRR amplitude. Here, we instead used a general linear mixed model (GLMM) to probe the relation between behavioral performance and response amplitude on the timescale of individual trials (*Chen et al., 2013*). We found that TRRs in early visual cortex scaled in amplitude with task difficulty, accuracy, reaction time (RT), and lapses, with decreasing strength ascending the visual cortical hierarchy. We found similar modulations in pupil size and cardiac activity. Importantly, task- and behavior-dependent modulations of the TRR were not explained by the artifactual influence of respiration and cardiac activity on cerebral blood oxygenation, which suggests that TRRs are linked directly to arousal. Our results demonstrate the existence of a widespread hemodynamic response in early visual cortex that reflects arousal on the timescale of individual trials. One intriguing possibility raised by our findings is that this arousal-linked response may alter encoding and/or decoding of visual information, thereby having a causal influence on behavior. For example, this might occur by modulating the cortical circuit computation to shift the balance in inference between prior and likelihood (*Dayan and Yu, 2006*; *Heeger, 2017*; *de Gee et al., 2020*).

## Results

### Task protocol and hypothesis

Human observers (*N* = 13) performed a visual orientation discrimination task consisting of separate easy and hard runs of trials while undergoing fMRI scanning (*Figure 1*). Each trial was 15 s, composed of a 0.2 s stimulus presentation and 14.8 s interstimulus interval. On runs of easy trials, the tilt of the stimulus (grating; diameter, 1.5°) was fixed at ±20° from vertical, yielding discrimination accuracy of ~90%. On runs of hard trials, the tilt was set adaptively with a staircase that yielded ~75% accuracy, typically converging to a tilt of ±1–4° from vertical. The fixation cross was a different color for easy

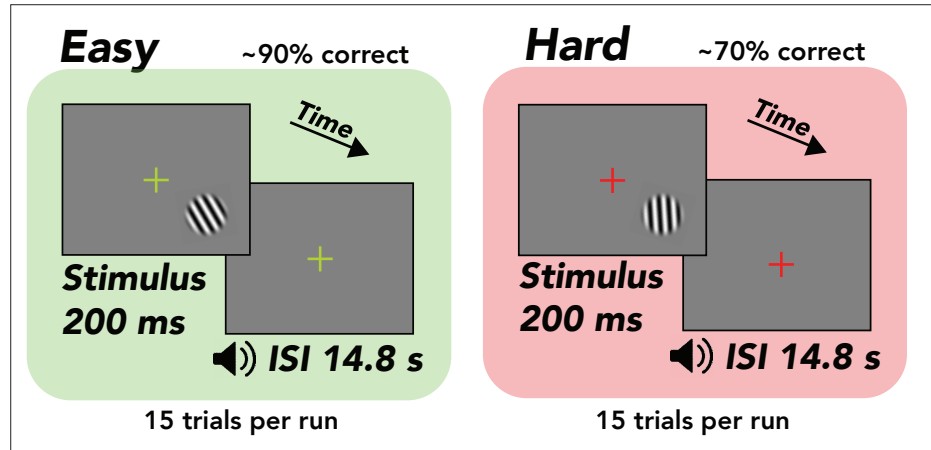

**Figure 1.** Task, orientation discrimination around vertical. Trial structure, 200 ms stimulus presentation, followed by a 14.8 s interstimulus interval (ISI), during which the observer made a button press response and immediately received tone feedback. Design, alternation between separate easy (~90% correct) and hard (~70% correct) runs comprising 15 trials each. Stimulus, tilted grating in a raised cosine aperture (diameter of 1.5°, but enlarged for illustrative purposes). Fixation cross, changed colors from green to red, indicating easy and hard runs, respectively.

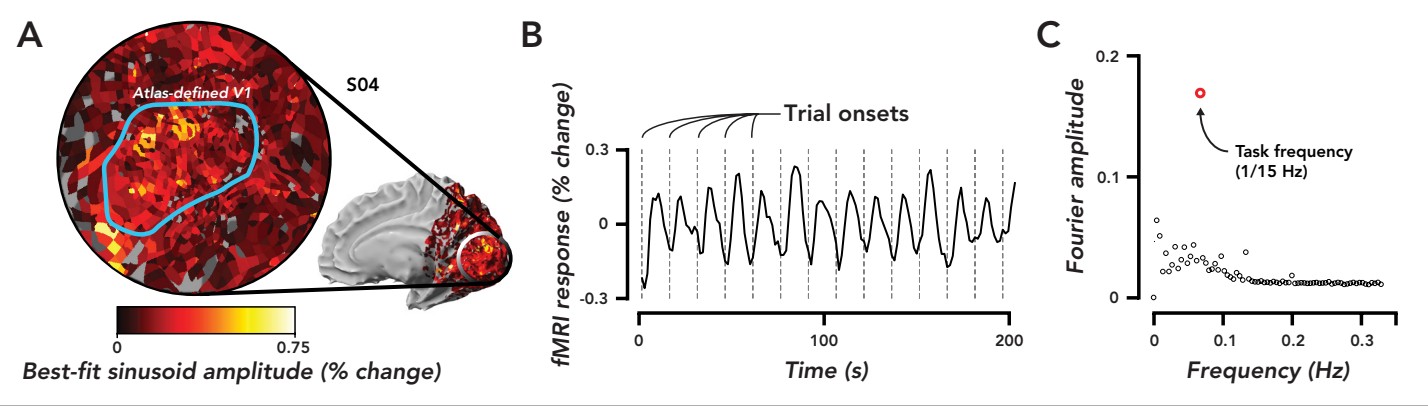

**Figure 2.** Widespread task-related fMRI responses in ipsilateral V1 were entrained to task timing. (**A**) Flat map centered on occipital pole showing fMRI responses. Inset, medial view of right hemisphere, indicated the region of cortex displayed in the flat map. Color map, amplitude of best-fitting sinusoid at the task frequency (1/15 Hz) in units of percent signal change for a single observer (O4). Blue Region of interest (ROI), atlas-defined V1, based on observer's anatomy. (**B**) Time course of fMRI responses averaged over all voxels in the V1 ROI defined in (A). Dotted gray lines, trial onsets. fMRI data are not artifact corrected in this figure. (**C**). Fourier transform of time course in (B). Open red circle, amplitude of the frequency response at the task frequency. Open black circles, the rest of the frequency response. The largest frequency component of the task-related response was at the task frequency.

and hard runs (green or red) to ensure that observers were aware of the difficulty of the task. The stimulus was always presented in the same location in the lower right hemifield such that stimulus-evoked responses and spatial attention responses were confined to a small region of the contralateral hemisphere. fMRI responses were measured in the right hemisphere, ipsilateral to the stimulus. These responses in the ipsilateral hemisphere were termed 'task-related' because they were neither stimulus evoked nor related to spatial attention (*Roth et al., 2020*). TRRs were estimated by averaging over all voxels on the cortical surface in ipsilateral V1. We first used linear regression to project out the global signal from each voxel in V1, a procedure that has been shown to be the most effective in reducing the influence of physiological variables on the fMRI signal (*Roth et al., 2020*; *Power et al., 2018*). To account for any possible residual physiological artifacts, we additionally predicted cardiac- and respiration-evoked hemodynamic activity (from cardiac and respiration signals measured during scanning) and projected these out of the fMRI data as well, a standard approach in the field (*Birn et al., 2008*; *Chang et al., 2009*). We used a GLMM (*Chen et al., 2013*) to assess the hypothesis that TRRs scale in amplitude with task difficulty and behavioral performance, while controlling for inter-observer differences in the fMRI signal.

## Task-related fMRI responses were spatially extensive and entrained to task timing

We observed extensive fMRI responses, extending past the boundaries of atlas-defined V1 (*Figure 2A*) and into other visual areas in the occipital, posterior parietal, and temporal cortex. We restricted the analysis to responses in ipsilateral V1, which corresponded to the hemifield in which no stimulus was presented (*Figure 2A*). These responses were entrained to trial timing (*Figure 2B*), i.e., the maximum amplitude component of the frequency response in ipsilateral V1 was 1/15 Hz, matching the frequency of the task (*Figure 2C*). Observers were instructed to covertly attend the peripheral stimulus, which was offset 5° from the central fixation cross. Thus, based on numerous prior studies (*Roth et al., 2020*; *Tootell et al., 1998*; *Gandhi et al., 1999*; *Herrmann et al., 2010*), it was assumed that the attention field was restricted to a small region in the contralateral hemisphere surrounding the location of the stimulus. The spatially extensive activity we observed suggests that TRRs were separate from stimulus-evoked and spatial attention responses.

## Head movement, cardiac activity, and respiration did not give rise to task-related fMRI responses

The influences of head movement, cardiac activity, and respiration on the fMRI signal were removed from the data using the following sequence: Multi-echo independent components analysis (ME-ICA)

(which reduces head movement artifacts [*Kundu et al., 2012*; *Kundu et al., 2013*; *Gonzalez-Castillo et al., 2016*]), global signal regression (*Roth et al., 2020*; *Power et al., 2018*), and projecting each movement/physiological regressor out of the resulting fMRI data. The maximum amplitude frequency component in the artifact-corrected time course (*Figure 3A*) remained 1/15 Hz (i.e. the task frequency), though the amplitude at this frequency was reduced slightly by the artifact correction. In the following sections, all analyses were performed on artifact-corrected time series.

## Task-related fMRI responses scaled with task difficulty

An observer needs to be alert to do well on a more challenging task. Therefore, we predicted that arousal would be higher on hard than easy trials, and that if TRRs reflect arousal, their amplitude would scale with task difficulty. Task-related responses in V1 were higher in amplitude on hard than easy trials (*Figure 3B*). To quantify this effect, we ran a GLMM which included the following predictors: the convolution of each observer's hemodynamic response function with impulses time-locked to (1) trial onset (which coincided with the stimulus presentation), (2) button press, and (3) a time-on-task boxcar in-between trial onset and button press; (4) task difficulty (easy or hard); (5) behavioral accuracy; (6) a prediction for the influence of cardiac activity (7) and respiration (8) on the fMRI signal following refs (*Birn et al., 2008*; *Chang et al., 2009*); (9) the translational and rotational movement of the head, consisting of six parameters (see Materials and methods). We also included terms that accounted for differences in the amplitude of the fMRI signal across observers, effectively normalizing the data across observers (*Chen et al., 2013*).

We focused our analysis on the interactions of task difficulty and accuracy with the linear combination of the three fMRI predictors (hereafter referred to as the 'combined fMRI predictor'), which were individually labeled fMRI_TO, fMRI_BP, and fMRI_ToT, corresponding to trial onset, button press, and time-on-task locked fMRI activity, respectively. It was essential to include in the model both difficulty and accuracy, as well as their interactions with each other and with the three fMRI predictors to isolate the effect of task difficulty from behavioral performance, which is also linked to arousal (*Burlingham et al., 2022*; *McGinley et al., 2015*; *de Gee et al., 2017*). The rationale is that we wanted to be as liberal as possible in our assumptions about which task events the TRR is responsive to, and also following similar models commonly used to link arousal to pupil size (*de Gee et al., 2014*; *Denison et al., 2020*) and task-related neural activity to the fMRI signal (*Yarkoni et al., 2009*) (see Discussion for more details). The full statistical results, with p-values for every predictor and interaction, are reported in the Supporting Information (*Source code 1*). The interaction of task difficulty with the combined fMRI predictor was significant ($p=1.85 \times 10^{-4}$, $F = 6.61$, $N = 16,399$; [fMRI_TO + fMRI_ToT + fMRI_BP]:difficulty), demonstrating that the amplitude of the TRR was larger when the task was more challenging.

## Task-related fMRI responses scaled with behavioral accuracy on individual trials

Task-related responses were higher in amplitude for trials on which observers made an incorrect behavioral report (*Figure 3B*). We hypothesized that the auditory feedback, which immediately followed the observer's button press and indicated whether the response was correct or incorrect, may have surprised observers on the infrequent incorrect trials, and hence increased arousal and thereby modulated the amplitude of the TRR (*Burlingham et al., 2022*). Indeed, the interaction of behavioral accuracy with the combined fMRI predictor was significant in the GLMM ($p=0.01$; $F = 3.81$; $N = 16,399$; [fMRI_TO + fMRI_ToT + fMRI_BP]:accuracy), as was their interaction with difficulty ($p=0.01$; $F = 3.79$; $N = 16,399$; [fMRI_TO + fMRI_ToT + fMRI_BP]:accuracy:difficulty). The amplitude of the TRR was highest on easy incorrect trials, when we expected that observers would be most surprised by their own (infrequent) errors (*Burlingham et al., 2022*).

## Task-related fMRI responses scaled with reaction time and this relation was captured by linear summation of three task-related components

Arousal influences behavioral performance, including RT (*Burlingham et al., 2022*; *Broadbent, 1971*), so we hypothesized that TRRs would differ according to RT. For correct trials, regardless of task difficulty, TRRs were higher in amplitude when the RT was fast (i.e. where 'fast' means above the median of the RT distribution for that specific trial type, e.g., easy/correct or hard/correct; *Figure 3C*;

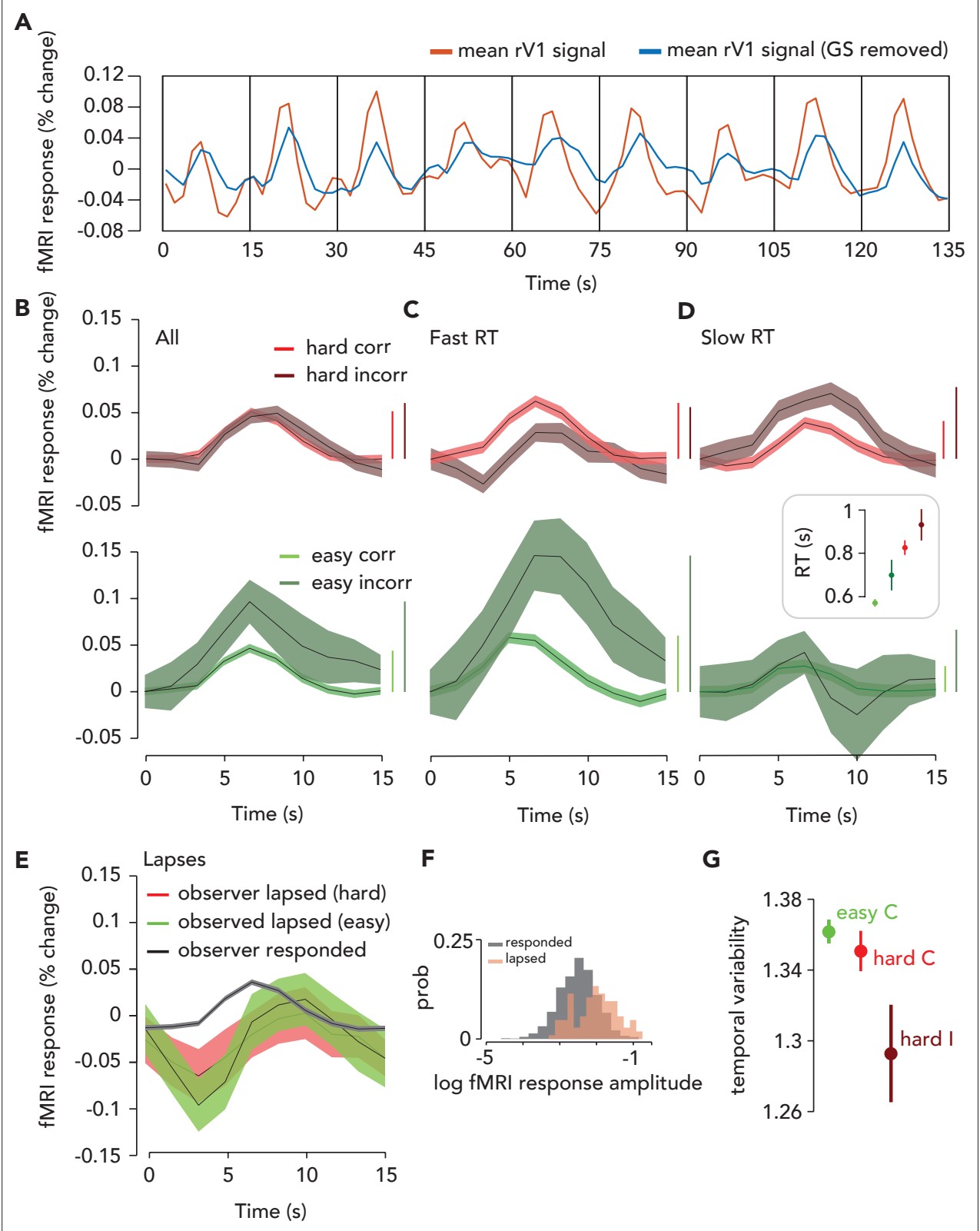

**Figure 3.** Task-related response (TRR) was modulated by task difficulty and behavioral performance. (**A**) To estimate the TRR, we projected the global signal ("GS," mean fMRI signal across all voxels in the brain) out of each voxel's fMRI response in ipsilateral (right) V1 (red, average across V1 voxels before correction). We then averaged over all voxels in ipsilateral V1 (blue). (**B**) TRR was modulated by task difficulty and behavioral accuracy. Average TRRs across trials and participants (*N* = 9) for hard and easy runs (red vs. green) and correct vs. incorrect trials (dark vs. light shades). Error surface, two

*Figure 3 continued on next page*

*Figure 3 continued*

SEM across observers. Vertical bars indicate response amplitude (max-min). (**C,D**) TRR was modulated by both accuracy and reaction time (RT). Same color scheme as (B), but (C and D) show only trials with either fast or slow RTs (i.e. below or above median of the RT distribution). Inset, mean RT for each trial type (errorbar, SEM). (**E**) TRR amplitude was modulated by lapses (missed response trials). Black curve, average TRR across all trials on which the observer made a button press response, irrespective of type. Red, lapsed trials on hard runs. Green, lapsed trials on easy runs. (**F**) Distribution of (log) fMRI amplitude (measured as std of the signal on each trial), across trials for lapse vs. response trials (orange vs. black). (**G**) Temporal variability of the TRR is modulated by difficulty and behavioral accuracy. Circle, mean temporal variability (measured as the circular std of the Fourier phase at the task frequency of 1/15 Hz) across observers ($N = 9$). Error bar, two SEM. Green, easy correct trials. Light red, hard correct trials. Dark red, hard incorrect trials. Easy incorrect not shown because it is far below the y-limit of the plot; extending the axes to include this data point would obscure differences between the other conditions (easy incorrect temporal variability: mean, 0.55; SEM, 0.22).

The online version of this article includes the following figure supplement(s) for figure 3:

**Figure supplement 1.** Task-related responses (TRRs) were modulated by reaction time (RT).

**Figure supplement 2.** Trial onset, button press, and time-on-task fMRI predictors in the general linear mixed model (GLMM), and their dependence on reaction time (RT).

**Figure supplement 3.** Amplitude modulation of the trial-averaged task-related response (TRR) with reaction time (RT) lies between two hypothetical extremes revealed by simulations.

*Figure 3—figure supplement 1*). For incorrect easy trials, this effect persisted and was even larger (*Figure 3D*; *Figure 3—figure supplement 1*), whereas for incorrect hard trials, the effect reversed and TRRs were higher in amplitude when RT was slow. Note that RTs also varied with task difficulty and behavioral accuracy, being fastest for easy/correct trials and faster for easy than for hard trials in general (*Figure 3D*, inset; GLMM with random intercept for each observer; difficulty: $F = 88.87$, $p<1\times10^{-6}$; accuracy: $F = 0.51$, p=0.47; difficulty X accuracy: $F = 4.7$, p=0.03).

RT was represented in our model via multiple components. We hypothesized that there are three inputs to the TRR: (1) a delta function at the trial onset time, (2) a delta function at the button press time, and (3) a boxcar extending between the trial onset and button press times (i.e. representing time on task; *Figure 3—figure supplement 2*). We hypothesized that amplitude modulation of the TRR with RT (on individual trials) was determined by the sum of these three hypothesized inputs. If this hypothesis is correct, the model and data should exhibit a similar pattern of amplitude modulation with RT on average across trials and participants. To test this, we performed a simulation in which we (1) predicted fMRI activity in response to the trial onset, button press, and time-on-task inputs using the RTs measured in the task, (2) averaged the simulated responses over all high and low RT trials (median split) and participants, crossed by difficulty and accuracy, and (3) compared the pattern of amplitudes between data and simulation. The predicted fMRI activity was computed by convolving a canonical HRF separately with each of the three inputs. We found that some aspects of the data were captured best by the time-on-task input ('fMRI_ToT') and other aspects were captured better by the linear sum of the trial onset and button press inputs ('fMRI_TO + fMRI_BP'), and overall the data lay somewhere in between these two extremes (*Figure 3—figure supplement 3*). We separated fMRI_ToT and fMRI_TO + fMRI_BP in the figure because we expected RT to have opposing effects on the amplitude of each. That is, for fMRI_TOT, a slower RT would result in a longer boxcar, which in turn would lead to a higher amplitude output. Conversely, for fMRI_TO + fMRI_BP, a slower RT would result in a larger separation between the trial onset and button press impulses, which would lead to a lower amplitude output (see Discussion for further comments). Specifically, the fMRI_TO + fMRI_BP simulation captured the pattern of amplitude modulation with RT on correct trials as well as the overall amplitude modulation with accuracy, and the fMRI_ToT simulation better captured RT-dependent amplitude modulation seen on incorrect trials. This validates our use of a model with a sum of the three inputs, which can interpolate between these two extremes in a flexible way. In conclusion, these results demonstrate that amplitude modulation of the TRR with RT may arise from linear summation of three task/behavior-related components, a model commonly used for linking arousal to pupil size (*de Gee et al., 2017*; *de Gee et al., 2014*; *Denison et al., 2020*; *van den Brink et al., 2016*).

## Task-related fMRI responses were large and delayed on lapsed trials

A previous study found that TRRs in macaque V1 were largest when the animal disengaged from the task (*Cardoso et al., 2019*), suggesting that TRRs may become largest, paradoxically, at very low arousal levels. We estimated TRRs in human V1 on lapse trials, defined as trials on which the observer

didn't respond. We found that TRRs were much larger in amplitude on lapse than response trials (*Figure 3E, F*; p=9.99x10⁻⁶, one-tailed permutation test), and this effect was stronger on easy than hard trials (hard trials: p=7.99x10⁻⁵; easy trials: p=9.99x10⁻⁶). TRRs were also delayed on lapse trials, peaking at 10 s after trial onset, rather than the typical 6 s on response trials. The TRR also displayed a more prominent decrease on lapse trials (*Figure 3E, F*).

## Task difficulty modulated trial-to-trial temporal variability in task-related fMRI responses

Reward magnitude is known to modulate the temporal variability, as well as amplitude, of TRRs (*Cardoso et al., 2019*; *Roth et al., 2020*). We tested whether task difficulty similarly modulates temporal variability. We measured temporal variability (across trials) in the TRR as the circular standard deviation of the Fourier phase at the task-onset frequency (1/15 Hz). The larger this number is, the more dispersed the response phases were across trials, with a maximum of square root of two if the phases were uniformly distributed, and a minimum of zero if the phases were identical across trials (a delta function). Temporal variability was modulated by both task difficulty and behavioral accuracy (*Figure 3G*), being lowest for easy incorrect trials, higher for hard incorrect, followed by hard correct, and highest for easy correct (compare with Figure 5 from *Roth et al., 2020*). For V1, V2, and V3, all pairwise comparison between these trial types was statistically significant (one-tailed permutation tests, $N = 9$) except the comparison between easy/correct and hard/correct trials, which trended in the same direction, but was not statistically significant (note that there were few data points [$N = 9$] for these specific comparisons). These results indicate that the modulation of temporal variability of the TRR is not specific to changes in reward (*Cardoso et al., 2019*; *Roth et al., 2020*), but rather may be observed in a range of task contexts in which arousal levels vary.

## Pupil size, respiration, and cardiac activity were similarly modulated by task difficulty and behavioral performance

If modulation of TRRs with task difficulty and behavioral performance reflect changes in arousal, we should observe that common physiological measures of arousal are also similarly modulated. The fMRI time series that we analyzed was corrected for artifacts caused by head movements, cardiac activity, and respiratory activity. Hence, the following analysis instead tests the idea that a common arousal process modulates both the TRR and physiological processes, such as cardiac and respiratory activity, independent on the artifactual impact they have on the BOLD signal. We had a group of observers ($N = 5$) perform the same behavioral task (with shorter 4 s trials) outside the fMRI scanner, while pupil size was measured with an infrared eye-tracker. The amplitude of the task-evoked pupil response was modulated by accuracy and task difficulty (*Figure 4A, B*; permutation test, p<0.05 for both comparisons, $N = 5$), consistent with previous studies (*Burlingham et al., 2022*; *Kahneman and Beatty, 1966*; *Kahneman and Beatty, 1967*; *Urai et al., 2017*; *Colizoli et al., 2018*). The observed modulation (incorrect > correct, with easy showing a bigger effect than hard) was similar to the fMRI results.

Heart rate, heart rate variability, and acceleration, as well as respiration volume, frequency, and variability (see Materials and methods for definitions), all of which are known to be influenced by arousal (i.e. via sympathetic and parasympathetic nervous system activity; see Materials and methods), also exhibited similar modulations (*Figure 4C–H*). We measured each of these variables during the fMRI experiment. We attempted a GLMM approach to predict the physiological measures from task and behavioral variables; however, standard checks of the residuals and fitted values indicated violations of the model's linear-Gaussian assumptions. Thus, we instead used non-parametric permutation tests to analyze dependency of the physiological measures on task difficulty and accuracy (see Materials and methods). We found that for runs of easy trials, all physiological measures except heart rate variability and heart rate acceleration were significantly modulated by accuracy (easy incorrect > easy correct; heart rate, p=0.032; heart rate variability, p=0.719; heart rate acceleration, p=0.507; respiration volume, p=0.004; respiration frequency, p=0.035; respiration variability, p=0.005). For runs of hard trials, only breathing frequency was significantly modulated by accuracy (hard incorrect < hard correct; p=0.015; comparisons for all other physiological measures: hard incorrect > hard correct; p>0.05). Only heart rate variability was significantly modulated by difficulty, analyzing correct trials only (hard correct < easy correct; p=0.005; comparisons for all other physiological measures: hard

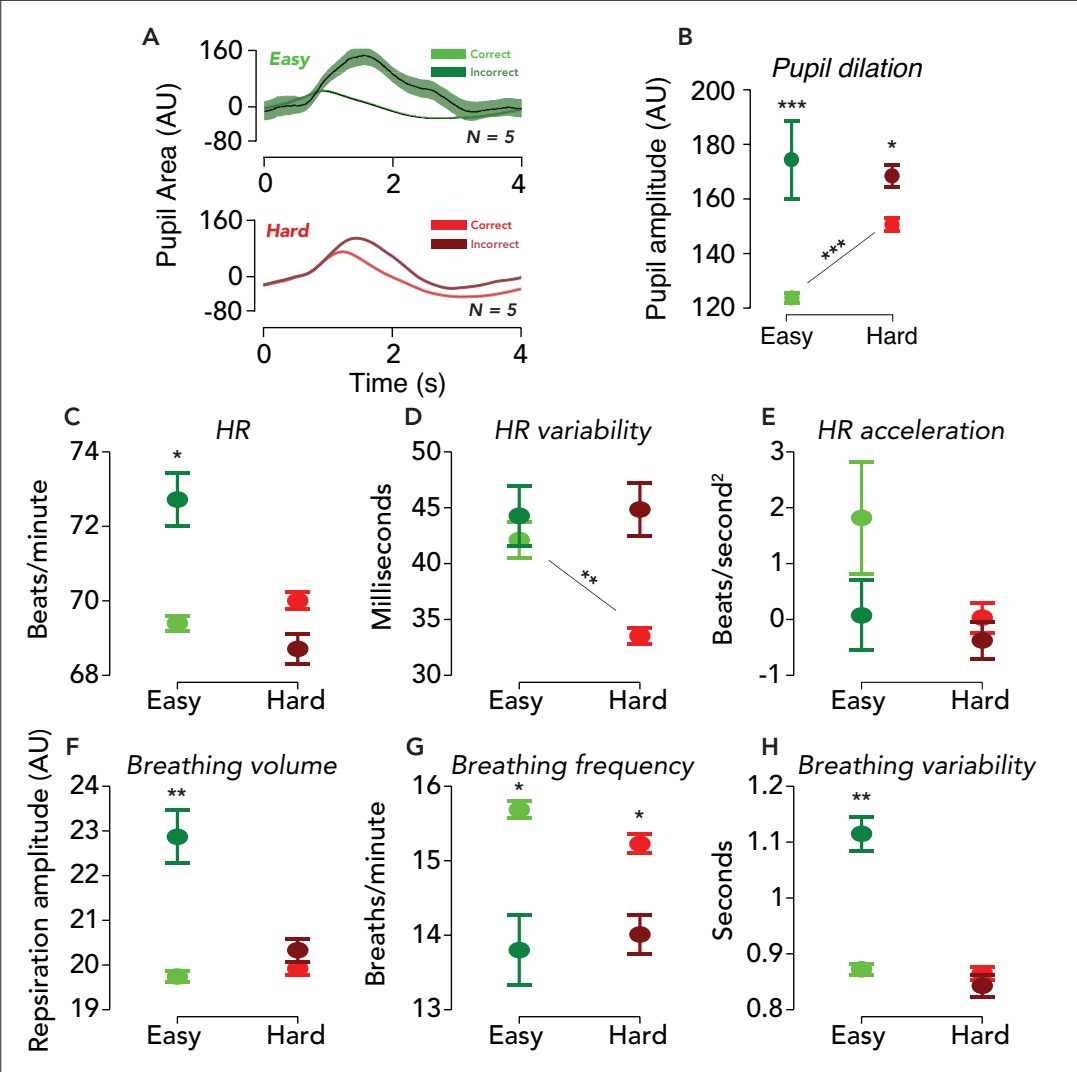

**Figure 4.** Physiological responses were similarly modulated by difficulty and behavioral accuracy. (**A**) Task-evoked pupil responses. Solid black lines, average task-evoked pupil response across trials and observers. Error surfaces, two SEM, across observers (*N* = 5). Green, easy runs of trials. Red, hard runs. Light colors, trials with a correct behavioral report. Dark colors, incorrect. All aspects of the pupil and fMRI protocols were identical, except the ISI was 3.8 s in the former and 14.8 s in the latter. (**B**) Amplitude of pupil dilation, measured as the standard deviation of the task-evoked pupil response time course. Error bar, two SEM. Asterisks, *p<0.05, **p<0.01, ***p<0.001 (permutation tests; one-tailed). Test statistic, the difference in the mean response amplitude for easy incorrect vs. correct trials, hard incorrect vs. correct trials, or hard correct vs. easy correct trials. (**C–H**) Physiological measurements collected during fMRI scanning were modulated by task difficulty and behavioral accuracy. Circle, average of physiological measure across time, trials, and participants. Error bar, two SEM (*N* = 9). Same format as (**B**), with asterisks indicating result of permutation test.

correct > easy correct; p>0.05). Physiological measures of arousal were modulated by task difficulty and behavioral performance in a similar manner to the TRR, suggesting arousal as a common driver.

## Task-related fMRI responses were progressively weaker throughout the visual cortical hierarchy

We measured TRRs in areas V1, V2, and V3, always ipsilateral to the visual stimulus. TRRs were highest in amplitude in V1, and became progressively weaker in V2 and V3 (*Figure 5A–C*). Amplitude modulation with difficulty and accuracy was found in each region, with the strongest modulations in V1, and becoming smaller in amplitude at progressively higher levels of the visual cortical hierarchy. TRR

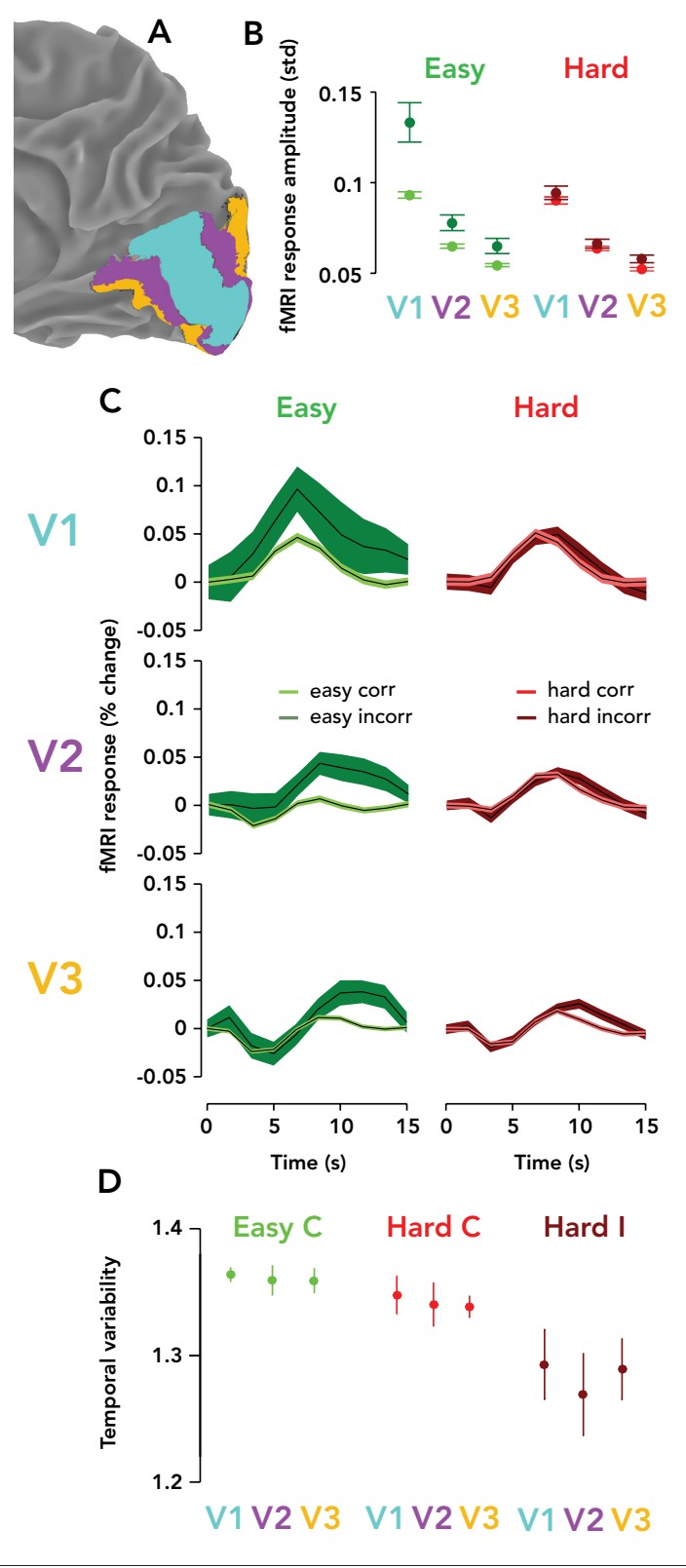

**Figure 5.** Task-related responses were progressively weaker ascending the visual cortical hierarchy. (**A**) Posterior third of the right hemisphere (ipsilateral to stimulus), Observer 8, with atlas-defined V1, V2, and V3 highlighted in three different colors (cyan, purple, orange, respectively). (**B**) fMRI response amplitude, measured as the standard deviation of the time course on each trial, scaled with task difficulty and behavioral accuracy in V1–V3. Weaker

*Figure 5 continued on next page*

*Figure 5 continued*

responses and modulations were observed in V3 than V2, and in V2 than V1. Circle, mean fMRI response amplitude across observers (*N* = 9). Error bar, two SEM. Green, easy runs. Red, hard runs. Light colors, correct trials. Dark colors, incorrect trials. (**C**) Mean TRRs across observers for different trial types and ROIs. Same format as *Figure 3B*. (**D**) Temporal variability is similar across V1, V2, and V3. Same format as *Figure 3G*. Easy incorrect not shown because it is far below the y-limit of the graph, and scaling the y-axis to show this datapoint would render the differences between the other conditions invisible (easy incorrect temporal variability, V1: mean, 0.55; SEM, 0.22; V2: mean, 0.56; SEM: 0.21; V3: mean, 0.53; SEM, 0.21).

The online version of this article includes the following figure supplement(s) for figure 5:

**Figure supplement 1.** Localizer and stimulus-evoked activity.

**Figure supplement 2.** fMRI responses in the left and right hemispheres.

amplitude was measured non-parametrically as the standard deviation of the response time course on a single trial and summarized visually in *Figure 5B*. For each trial type (easy correct, hard correct, easy incorrect, hard incorrect), each pairwise difference between V1/2/3 TRR amplitudes was statistically significant ($p < 0.05$; one-tailed permutation tests, V1 >V2 > V3) except for two non-significant comparisons: easy/incorrect V2 >V3 and hard/incorrect V2 >V3. The temporal variability of the TRR, on the other hand, was similar between V1, V2, and V3 (*Figure 5D*; $p > 0.05$ for all permutation tests). The shape of the TRR was also different in V2/3 vs. V1 — being slightly more delayed and with a larger initial dip that sometimes went negative. Interestingly, these dynamics were more similar to those seen on lapsed trials in V1 (*Figure 3E*).

## An eye-movement-evoked artifact could not account for ask-related fMRI responses

One might imagine an artifactual explanation of TRRs that goes like this: when the eyes move, this causes large changes in retinal stimulation where visible elements of the environment, such as the screen edge, move on the retina. This retinal stimulation might, in turn, evoke visual responses in V1. The scanner room was dark, but the screen edge was visible, so we might expect eye movement–related artifacts in V1 at the retinotopic position corresponding to the screen's edge. In some conditions (e.g. when the task is easier and there is an error), observers might move their eyes more, causing larger visual responses in V1, thereby 'explaining away' our findings. We ruled out this possibility in two ways. First, we analyzed fixation stability during the task (measured outside the scanner in a separate pupillometry session) and tested for systematic differences in fixation stability between conditions (difficulty, accuracy, RT). We quantified fixation stability as the variability (std) over time within a trial of the distance between gaze and fixation in degrees of visual angle. Fixation stability (median across trials) was 0.18° on average across participants (for each observer: 0.18, 0.21, 0.20, 0.15, 0.17°), i.e., each observer fixated stably on the central cross. We performed a GLMM fitting fixation stability across trials with difficulty, accuracy, RT, and their interactions, as well as random intercepts for each subject and session. There were no statistically significant effects (all $p > 0.4$, all $F < 0.34$), i.e., no evidence to suggest that fixation stability varied between conditions.

Next, we tested whether fMRI activity in V1 was highest at the retinotopic locations corresponding to the screen edge, as predicted by this luminance-artifact hypothesis. Specifically, we measured the average BOLD amplitude across all visual field locations within a rectangular border with a 'thickness' of 7.2° surrounding the screen edge (which extended from –35 to 35° horizontally and from –19.75 to 19.75° vertically), derived from the visual field amplitude maps (*Figure 5—figure supplement 1*). We compared this test statistic to a null distribution comprising the average BOLD amplitude within 100,000 randomly selected rectangular borders in the visual field, with the same visual area in degrees squared. These random borders' centers were drawn from a uniform distribution. We computed an exact p-value as the proportion of the null distribution beyond the test statistic (right-tailed) and found no evidence of a statistically significant difference in amplitude for easy ($p = 0.39$, *N* = 100,000 samples) or hard runs of trials ($p = 0.56$, *N* = 100,000 samples). The statistical outcome ($0.5 < p < 0.7$) was the same regardless of the border thickness (varied between 1.8 and 10.8°). This demonstrates that visual responses arising from the screen edge and eye movements, if present, do not account for the TRR.

## Discussion

We observed large fMRI responses in human early visual cortex that were spatially widespread, independent of visual stimulation, and tightly entrained to trial timing. These TRRs, measured in V1 in the hemisphere ipsilateral to the stimulus, were larger in amplitude when the task was more difficult, and on trials on which observers made an erroneous response, failed to respond, or were faster to respond. Neither the TRR nor its modulation with task difficulty or behavioral performance was explained by common fMRI-BOLD artifacts, i.e., changes in head movement, cardiac activity, and respiration, nor by the fMRI 'global signal'. The temporal variability of the TRR was also modulated by task difficulty and behavioral performance, complementing similar findings from previous studies that manipulated reward (*Cardoso et al., 2019*; *Roth et al., 2020*). Contrary to the consensus of opinion that the fMRI-BOLD signal in V1 primarily reflects stimulus-evoked and attentional neuronal activity, we found evidence that the BOLD signal also contains a component that reflects the demands of a task and behavioral performance on the timescale of individual trials.

TRRs in early visual cortex are modulated by task difficulty, reward, and behavioral performance. Two previous studies *Cardoso et al., 2019* and *Roth et al., 2020* examined the effect of reward magnitude on TRRs in early visual cortex, and found that larger rewards led to decreased temporal variability and increased response amplitude on individual trials, as well as larger trial-averaged response amplitude. *Roth et al., 2020* found that multiple measures of trial-to-trial variability in TRRs (i.e. timepoint, amplitude, and temporal variability) changed with reward. The authors used computational simulations to show that simultaneous increases in trial-averaged TRR amplitude with reward and decreases in the three measures of variability were consistent with only one possibility: changes in trial-to-trial temporal jitter. That is, averaging over many temporally jittered responses led to a lower amplitude trial-averaged response whereas averaging over many temporally aligned responses led to the opposite. And likewise, more temporal jitter also generated increases in all measures of trial-to-trial variability (timepoint, amplitude, and temporal variability). However, these results don't preclude additional changes in trial-to-trial response amplitude with reward, but instead suggest that a portion of the trial-averaged response amplitude cannot be predicted by trial-to-trial amplitude alone. Indeed, Cardoso et al. found that both the trial-to-trial amplitude and timing of TRRs in macaque V1 were modulated by reward (*Cardoso et al., 2019*). Our results provide a complementary account. We found that trial-to-trial amplitude, trial-averaged amplitude, and temporal variability of the TRR were all modulated by task difficulty and behavioral performance. Temporal variability was reduced for unexpected task events (i.e. incorrect feedback tone on an easy run of trials), consistent with previous findings showing that temporal precision (1/variability) scaled with arousal (*Cardoso et al., 2019*; *Roth et al., 2020*). Taken together, the evidence suggests that reward, task difficulty, and behavioral performance modulate both the temporal variability and amplitude of TRRs.

We found that TRR amplitude was modulated by RT on a trial-to-trial basis. It is well known that arousal influences RTs, with a classic inverted-U relation (*Widdicombe and Sterling, 1970*), an effect that shows up in standard physiological measures of arousal, including pupil size (*Burlingham et al., 2022*) and cardiac activity (*Chin and Kales, 2019*). So if TRR amplitude reflects arousal, as we hypothesized, we would expect that it would be higher when an observer responds faster, which is what we found (on correct trials). Note that we assume that our task primarily operates in the left half of the inverted-U, because even the hard task is rather boring, so arousal levels are never very high (*Burlingham et al., 2022*). Previous fMRI studies offer an alternative, but complementary perspective on the relationship between RT and TRR amplitude. A previous study measured the TRR in human V1 in a visual detection task (*Jack et al., 2006*) with either an immediate or delayed behavioral report (delay, ~9 s). In the delayed condition, there were response peaks following both trial onset *and* the behavioral report. In peripheral V1, far from the representation of the stimulus, the amplitude of fMRI activity was lower for a delayed than immediate report. This indicates that for an immediate report, there were also two components of the TRR — one for the trial onset and one for the response — but because they were so close in time, their impulse responses summed to a unimodal response. We might assume that two such 'blurred-together' responses were also present in our experiment, in which observers responded immediately after the stimulus presentation. It would follow then, that differences in RT should affect this summation and thereby modulate the amplitude of the unimodal TRR — slower RTs associated with lower fMRI amplitudes (*Figure 3—figure supplement 2A*, *Figure 3—figure supplement 3B*). Conversely, it could be that the amplitude of the TRR

reflects time-on-task, i.e., a sustained input between trial onset and RT, often modeled as a boxcar (*Yarkoni et al., 2009*). If true, TRR amplitude would be higher for trials with slower RTs (*Figure 3— figure supplement 2B*, *Figure 3—figure supplement 3C*). And a third possibility is that all three inputs are present, at trial onset, at the response time, and time on task. This scenario would produce some effect in-between these two extremes, which depends flexibly on the relative strengths of the three inputs. A model similar to this, with three inputs, is commonly used to predict the task-evoked pupil response from task structure (*Burlingham et al., 2022*; *de Gee et al., 2014*; *Denison et al., 2020*). To be liberal in our assumptions, we used a GLMM with these three putative inputs — fMRI_TO, fMRI_BP, and fMRI_ToT — as well as their combinations. We found that the combination of all three inputs best explained the results (i.e. the interactions of the combined fMRI predictor with task and behavioral variables were significant in the model). This agrees with the results of many studies linking arousal and pupil size, which used a similar model (*de Gee et al., 2017*; *de Gee et al., 2014*; *Denison et al., 2020*; *van den Brink et al., 2016*). Differences in RT are associated with changes in the timing and variability of phasic locus coeruleus (LC) activity (*Aston-Jones and Cohen, 2005*). And pupil size changes are linked closely to changes in neural activity in LC (*Joshi et al., 2016*) as well as other neuromodulatory brainstem loci, including the dorsal raphé (*Cazettes et al., 2021*), ventral tegmental area, subsantia nigra, and basal forebrain, and other subcortical and cortical structures (*Burlingham et al., 2022*; *de Gee et al., 2017*; *Joshi et al., 2016*). Thus, our results suggest that the link between TRR amplitude and RT is mediated by neuromodulatory influences on visual cortex and/ or other, decision-related areas of the brain.

We found that TRRs and pupil responses were both modulated by task difficulty and behavioral accuracy in a similar manner. Previous empirical and computational modeling work suggests that such modulations may emerge via pupil-linked arousal systems being driven by internal beliefs about ones own behavioral choices (*Urai et al., 2017*; *Colizoli et al., 2018*). Signal detection theoretic models that take into account ones uncertainty about ones own choices (i.e. confidence) correctly predict modulations of the task-evoked pupil response before and after external feedback and their scaling with difficulty and accuracy (*Urai et al., 2017*; *Colizoli et al., 2018*). According to these models, the brain might use its arousal system to 'broadcast' these uncertainty signals to a wide set of brain regions in order to facilitate inference and information processing. Under this account, the TRR's modulation with difficulty and behavior may reflect adaptations in an underlying computational process to task demands, internal sensory noise, and internal beliefs (decision uncertainty). Accordingly, we found modulations of the task-evoked pupil response and TRR with difficulty and accuracy that matched predictions from this model (compare our *Figures 4B and 5B* with Figure 2B from *Colizoli et al., 2018* ). This decision-uncertainty-arousal model also predicts that RTs vary in the same manner (i.e. fastest to slowest: easy correct, hard correct, hard incorrect, easy incorrect) — however, our findings weren't consistent with this. Only 1/9 observers showed the expected pattern of RTs; instead, RTs were faster overall for easy trials, regardless of accuracy (*Figure 3D*, inset). This difference may be due to the fact that there was no variability in stimulus orientation in our easy task, but there was in the hard task (due to the staircase), whereas the stimulus variability was the same across difficulty levels in refs (*Urai et al., 2017*; *Colizoli et al., 2018*). A ripe area for future study would be to link decision-theoretic models such as this with a linear systems model of the fMRI signal, and try to predict the TRR from task and behavioral parameters.

Our results are not attributable to the effects of spatial attention. Previous studies have measured the spatial distribution of spatial attention (the 'attention field' ) in early visual cortex and found that it tightly surrounds the retinotopic location of the attended stimulus (*Herrmann et al., 2010*; *Carandini and Heeger, 2011*), particularly when it appears at a predictable location, size, and time, as in our experiment. Furthermore, we previously found, using a similar stimulus and protocol, that reward magnitude modulated TRR amplitude (also measured in ipsilateral cortex) even though behavioral accuracy and RT were statistically indistinguishable between low and high reward conditions (*Roth et al., 2020*). That is, increased reward did not alter the allocation of spatial attention, as quantified as a behavioral enhancement, which is how attention is typically operationalized (*Dugué et al., 2020*). In our study, we found that the measured TRR was lower in amplitude ascending the visual cortical hierarchy. This is opposite to what is seen in fMRI studies of spatial attention responses, which increase in amplitude from V1 to V2 to V3 (*Dugué et al., 2020*).

The origin and control of the TRR in early visual cortex remain unclear. One possibility is that they are driven by an intrinsic vascular mechanism rather than a local neuronal one (*Sirotin and Das, 2009*; *Cardoso et al., 2012*; *Cardoso et al., 2019*; *Bekar et al., 2012*; *Sirotin et al., 2012*; *Herman et al., 2017*; *Das et al., 2021*). This putative mechanism, vasomotion, or oscillation in tone of blood vessels independent of cardiac and respiratory activity, occurs in cerebral arteries and is influenced by arousal (*Nilsson and Aalkjaer, 2003*; *Borovik et al., 2005*; *Mateo et al., 2017*). More broadly, cerebral blood flow is complex and modulated by arousal via multiple cellular mechanisms (*Bekar et al., 2012*; *Raichle et al., 1975*), including activity of astrocytes (*Duffy and MacVicar, 1995*; *Bekar et al., 2008*; *Takata et al., 2015*; *Wang et al., 2018*), pericytes (*Peppiatt et al., 2006*; *Hall et al., 2014*), endothelial cells (*Sorriento et al., 2011*), and smooth muscle cells (*Faber, 1988*). Our fMRI data cannot reveal the origins of the TRR, but may provide some clues. We found that the amplitude of the trial-averaged TRR was highest in V1, and progressively smaller in V2 and V3 (*Figure 5A*), whereas temporal variability was similar among the three visual areas (*Figure 5D*). On the other hand, both temporal variability and response amplitude were modulated by task difficulty and behavioral performance *within* each visual area (*Figure 5A, D*). The larger TRRs in V1 (vs. V2 and V3) may arise from some non-neural factor, possibly the unique vasculature of V1, where there are two input arteries, the calcarine artery (branch of the posterior cerebral artery) and posterior temporal artery (branch of the middle cerebral artery). This would support the view that the TRR is purely vascular in origin (*Das et al., 2021*) and may be modulated (via vascular mechanisms) by neuromodulatory activity arising in the brainstem. It's also possible that the TRR is actually not smaller across the visual hierarchy, but instead simply appears so because of how the fMRI signal pools across local neural and/or vascular processes. Furthermore, it's known that some neuromodulatory projections are stronger in some areas than others, meaning that the pooled magnitude of the TRR might be higher in one area than another, even if the local process and its function is the same. For example, the density of noradrenergic (*Foote and Morrison, 1987*; *Shimegi et al., 2016*) and cholinergic (*Disney et al., 2006*) projections differs in V1 and V2.

The TRR may be neural in origin. Our pupillometry results suggest that arousal increases at trial onset in our task. It's well known that the noradrenergic projections are relatively unspecific (*Foote and Morrison, 1987*; *Aston-Jones and Waterhouse, 2016*). If the NE-arousal brainstem system is active at task onset, this would be expected to modulate neural activity in early visual cortex, given prior anatomical and physiological studies (*Foote and Morrison, 1987*). A number of previous studies have proposed that neuromodulators can alter the weighing of new evidence and prior expectations in cortical computations via changes in synaptic gain (*Dayan and Yu, 2006*; *Heeger, 2017*; *de Gee et al., 2020*; *Friston, 2010*; *Moran et al., 2013*), with NE tamping down the effect of the prior. Consistent with this idea, pupil-linked arousal has been found to primarily affect decision biases, which can be modeled as a change in the weighing of prior vs. likelihood. For example, *de Gee et al., 2017* and *de Gee et al., 2020* found that larger pupil responses were linked with reductions in decision bias (measured via the signal detection theoretical criterion) in a visual detection task and increased activity in a number of neuromodulatory loci, measured using fMRI. Another study manipulated prior expectations on a trial-by-trial basis and found that pupil size dynamically tracked resulting decision biases in human behavior, with smaller pupil responses corresponding to smaller biases (*Krishnamurthy et al., 2017*). This, taken together with other results showing that pupil-linked arousal also varies with serial choice biases, decision confidence, and prediction errors (*Urai et al., 2017*; *Colizoli et al., 2018*), suggests that a portion of variability in behavioral choices may be causally controlled by arousal. Our results, considered from this perspective, may indicate that TRRs have a computational role, shaping neural activity and/or readouts in visual cortex and thereby having a causal effect on behavior. The TRRs we measure with fMRI may therefore reflect the widespread effect of neuromodulators like NE on cortical computation via normal neurovascular coupling (*McGinley et al., 2015*), separate effects of NE or other neuromodulators on the vasculature (*Bekar et al., 2012*; *Purkayastha and Raven, 2011*), or a combination of these factors.

Our findings may also be related to recent neurophysiology studies in animal models linking arousal, global brain states, and visual cortical activity. New recording techniques have allowed researchers to measure the activity of many thousands of neurons at once (*Musall et al., 2019*; *Steinmetz et al., 2019*; *Stringer et al., 2019*). Studies using such techniques have observed widespread responses in visual cortex (and beyond) during delay period and behavioral response times, i.e., in the absence of a stimulus. A number of studies have found that arousal, measured by pupil size, explains much of

the variability in this widespread visual cortical activity (*Steinmetz et al., 2019*; *Stringer et al., 2019*; *Salkoff et al., 2020*). Furthermore, in mouse V1, hemodynamic responses recorded during visual stimulation can be decomposed into a local and global component, the latter of which is correlated with changes in pupil size, consistent with the hypothesis that the global component reflects arousal (*Pisauro et al., 2016*). It remains to be seen if these widespread neural responses give rise to the TRR. We found that the temporal precision of the TRR across trials scaled with arousal level. This is consistent with the hypothesis that the TRR is neural in origin, given previous findings demonstrating that increased alertness leads to more precise neural activity in visual cortex (*McGinley et al., 2015*; *Lombardo et al., 2018*; *Arazi et al., 2019*). If the TRR is neural in origin, it may be driven by a relatively small subpopulation of neurons, not evident in LFPs or mean firing rates (*Sirotin and Das, 2009*; *Das et al., 2021*), or by widespread changes in subthreshold neuronal activity (*McGinley et al., 2015*). The TRR may instead, as described above, be evoked by a widespread release of neuromodulators that both evoke a hemodynamic response and shift the weighing of prior vs. likelihood, both of which may occur without a change in mean firing rates or mean LFP amplitude.

A previous study, *Cardoso et al., 2019*, found that macaque V1 exhibited alternating increasing and decreasing ramps in blood volume on the timescale of minutes between blocks of low and high reward. This is an interesting result that suggests that the TRR is characterized by both differences in the baseline blood volume and the amplitude of hemodynamic responses. Unfortunately, unlike optical imaging, fMRI-BOLD is not sensitive to such slow changes in the baseline, because the signal is affected by many other sources of noise in the same slow frequency range. Future studies using alternative imaging methods like Vascular-Space-Occupancy (VASO) could potentially resolve these slow ramps in blood volume and validate the results of ref (*Cardoso et al., 2019*) in humans.

Our findings may be related to previous studies identifying a component of the fMRI-BOLD signal related to arousal, but further research is necessary to test if this component and the TRR reflect the same phenomenon (*Chang et al., 2016*; *Özbay et al., 2019*; *Gonzalez-Castillo et al., 2021*; *Goodale et al., 2021*). One of these studies *Goodale et al., 2021* found that the most responsive voxels measured were in the primary sensory cortices, including early visual cortex, and that these voxels alone sufficed for prediction of behavioral performance and an EEG measure of arousal. If future studies find that this fMRI component is equivalent to the TRR, it could lead to a productive unification of these two areas of research.

The amplitude of the TRR in V1 (mean across all voxels and trials, 0.32% change; median, 0.28% change; max, 6.20% change) is similar to that of stimulus-evoked and attentional hemodynamic responses (*Cardoso et al., 2012*; *Roth et al., 2020*; *Dugué et al., 2020*; *Das et al., 2021*; *Lewis et al., 2018*). This raises concerns about fMRI task designs in which attention or other psychological variables are operationalized according to behavioral performance and/or task parameters. In such cases, TRRs, which are modulated by critical task variables and behavior, may mix with other fMRI signal components, and thereby confound analysis. Our results demonstrate that common preprocessing approaches (i.e. global signal regression, physiological 'noise' correction) do not remove the TRR from fMRI data. Furthermore, the TRR contains information about brain state that would otherwise typically be measured independently with pupil size and/or cardiac activity. Therefore, in some scenarios it may be useful to model the TRR, remove it from fMRI data as a preprocessing step, and then possibly analyze it as an independent measure of arousal. One way to achieve this is to run a 5–10 min version of a task in which the visual stimulus is confined to one visual field, so that the TRR can be easily isolated and measured later (see *Donner et al., 2008* for a simple protocol for removing the TRR).

## Conclusion

Approaches to analyzing and interpreting fMRI data are largely based on studies of early visual cortex, where neural activity is time-locked to visual stimulation, and the fMRI signal is well approximated by a linear transformation of that stimulus-evoked neural activity (*Ress et al., 2000*; *Heeger and Ress, 2002*). However, it has been known for over a decade that the fMRI signal in early visual cortex contains large components not predicted by local spiking or visual stimuli (*Sirotin and Das, 2009*; *Ress et al., 2000*; *Jack et al., 2006*). One of these components, the task-related fMRI response, is widespread and entrained to task timing, and it covaries with physiological measures of arousal as well as reward magnitude (*Sirotin and Das, 2009*; *Cardoso et al., 2012*; *Cardoso et al., 2019*; *Roth*

*et al., 2020*; *Sirotin et al., 2012*; *Herman et al., 2017*). In this study, we found that the TRR is also modulated across trials by task difficulty, accuracy, RT, and lapses. These findings demonstrate the existence of a component of the fMRI-BOLD signal in human early visual cortex that reflects arousal on the timescale of individual trials.

## Materials and methods

### Observers

Experiments were conducted at two sites. fMRI, cardiac, and respiration measurements were acquired from 13 observers at the Functional Magnetic Resonance Imaging Core Facility at the National Institutes of Health (NIH). Pupil data were acquired from five other observers (two males, three females) at New York University (NYU). All observers were healthy adults, with no history of neurological disorders and with normal or corrected-to-normal vision. For the fMRI experiment, four observers were missing either high-quality cardiac and/or respiratory data, so all of their data was excluded from all analyses. Experiments were conducted with the written consent of each observer. The consent and experimental protocol were in compliance with the safety guidelines for MRI research and were approved by both the University Committee on Activities involving Human Subjects at New York University (RB-FY2016-158), and the Institutional Review Board at the National Institutes of Health (93M-0170, https://clinicaltrials.gov/ identifier: NCT00001360).

### Experimental protocol

Observers performed a two-alternative forced choice orientation discrimination task, reporting whether a stimulus was tilted clockwise or counter-clockwise relative to vertical (*Figure 1*). The stimulus was a grating patch (spatial frequency, 4 cycles/°) multiplied by a circular envelope (diameter, 1.5°) with raised-cosine edges (contrast, 100%). The stimulus was equiluminant with the gray background, such that it would not evoke a luminance response in visual cortex or a light reflex response in the pupil (*Burlingham et al., 2022*). On each trial, the grating was flashed briefly in the lower right visual field (5° diagonal to fixation) and observers covertly attended to the stimulus. Stimulus location was determined by the recording chambers used in related monkey electrophysiological experiments (*Cardoso et al., 2019*), which are typically centered at ~4–5° eccentricity in the lower visual hemifield, along the diagonal, i.e., ~45 or 135° polar angle. The difficulty of the task was manipulated by changing the tilt angle of the grating. In blocks of easy trials, the grating was tilted ±20° away from vertical. In blocks of hard trials, the grating was tilted by a much smaller amount (typically ±1°), with an adaptive staircase (one-up, two-down) ensuring ~70–75% correct discrimination accuracy. Observers were instructed to fixate a central cross on the monitor throughout the experimental session. The color of the fixation cross indicated the difficulty of the current block of trials — red for hard and green for easy blocks. This ensured that the observer was always informed of the current task difficulty. Each block had a predictable trial structure with a fixed interstimulus interval. Each trial was 15 s long, starting with a 200 ms stimulus presentation and a 14.8 s interstimulus interval, during which the observer was instructed to make a key press response indicating the perceived orientation of the grating. Observers could respond at any time during the ISI. 99.6% of RTs were under 4 s, and 86.2% were under 1 s (median RT: 552 ms). A long ISI was employed because it allowed us to better measure the full-time course of the TRR. Tone feedback was given immediately following the behavioral response — a high tone indicated correct behavioral responses and a low tone indicated incorrect behavioral responses. There were 16 trials per run (240 s), but the first trial was always removed in our analysis to allow the hemodynamic response to reach steady state. Stimuli were generated using Matlab (MathWorks, MA) and MGL (*Gardner, 2018*) on a Macintosh computer. Stimuli were displayed via an LCD screen. Observers viewed the display through an angled mirror (field of view: 70 × 39.5°).

### fMRI data acquisition

Blood oxygenation was measured in occipital, parietal, and temporal cortex with fMRI (3T GE scanner, 32-ch coil, multiecho pulse sequence with echo times of 14.2 ms, 30.1 ms, and 46 ms; 22 slices covering the posterior third of the brain; voxel size 3 × 3 × 3 mm; TR of 1.5 s). fMRI time series were motion corrected (*Nestares and Heeger, 2000*), and then the time series from the three echos were combined into a single time series using ME-ICA (*Kundu et al., 2013*).

## fMRI analysis

fMRI data were aligned with base anatomical scans from each observer using a robust image registration algorithm (*Nestares and Heeger, 2000*). The boundaries of visual areas V1, V2, and V3 were defined using an anatomical template of retinotopy (*Benson and Winawer, 2018*), and then projected onto each individual volumetric data, so as to avoid any interpolation or smoothing of the fMRI time series data (*Fischl, 2012*). To characterize the TRR in ipsilateral V1, we computed the amplitude and phase of the sinusoid at the task frequency (1/15 Hz), which best fit the fMRI time series. Fourier analysis was used as validation that the largest frequency component of the TRR was at the task frequency. For *Figure 5A*, the fMRI response amplitude (and pupil response amplitude) was computed as the standard deviation of the response time course over a single trial.

## Cardiac, respiratory, and head movement signals

While in the scanner, cardiac activity was measured using a pulse oximeter (GE Medical Systems E8819EH; sampling rate, 50 Hz), respiration was measured using a respiration belt (GE Medical Systems E8811ED; sampling rate, 50 Hz), and head movements were tracked continuously using real-time head motion estimates implemented in AFNI (*Cox, 1996*). To obtain a precise estimate of the peak times in the cardiac signal, given the temporal undersampling of the pulse oximeter, we used linear extrapolation based on signal to the left and the right of each putative undersampled peak location and solved for the missing peak as the intersection of the two best-fit lines. Heart rate was computed as the mean of the time series of interpeak durations and converted into beats per minute. Heart rate variability was computed using the root mean square of the successive differences (RMSSD) method (*Shaffer and Ginsberg, 2017*), which acts as a high-pass filter, removing low frequency components of the time series of interpeak durations, such as those related to respiration. Heart rate variability is a widely used measure of parasympathetic nervous system activity (*Goldberger et al., 2001*) and is typically higher when a participant is more calm. Heart rate acceleration was computed as the mean of the first derivative of the time series of interpeak durations, with units of beats/s$^2$, so positive numbers reflect acceleration and negative numbers deceleration. Acceleration/deceleration of the heart beat is associated with changes in either parasympathetic (vagus nerve) or sympathetic nervous system activity (*Pan et al., 2016*). Likewise, there is both parasympathetic (vagus nerve) and sympathetic innervation of the lungs, and thus respiration can reflect arousal via autonomic nervous system activity (*Widdicombe and Sterling, 1970*). The respiration signal was mean-subtracted and converted into percent signal change. Its amplitude (volume) was obtained by taking the standard deviation of the time series on each trial separately. Respiration frequency was computed, on each trial, as the component with the maximum amplitude in the frequency response. People hold their breath more when more alert, leading us to predict that respiration frequency should be lower when arousal is higher (*Widdicombe and Sterling, 1970*). Respiration variability was computed in the same way as heart rate variability, without the peak extrapolation method, by finding the signal peak times and computing the RMSSD thereof. We included all of these physiological measures because they capture different components of arousal (e.g. reflecting sympathetic and/or parasympathetic nervous system activity), which may have different links with behavior and task demands. Head movement estimates were computed from the times series from the first TE of the multiecho sequence, and then this estimate was used to motion correct the images from all three TEs (*Kundu et al., 2013*). We used these estimates of the translational and rotational movement of the head in each dimension of movement (roll, pitch, yaw, y, x, and z) as predictors in the GLMM. All physiological measures were computed with a time bin of 15 s (the duration of a trial).

To analyze the influence of task difficulty and behavioral accuracy on physiological measures of arousal, we binned these physiological measures by trial type (difficulty × accuracy) and ran permutation tests to check for differences among trial types. Permutation tests were performed to assess the differences in the amplitude of task-evoked pupil responses as well. We computed a test statistic from the data, the difference in the mean physiological measure across trials between easy correct and easy incorrect trials, hard correct and hard incorrect trials, and easy correct and hard correct trials. For heart rate variability and breathing frequency only, the direction of the test was reversed, as we expected these measures to be lower when arousal is greater. We constructed a null distribution by concatenating the data from the two conditions, randomly shuffling the trial type labels (10,000 iterations), splitting the resulting array into two new groups (with the same sizes as the original two labeled

groups), and computing the difference of the mean of these two new groups. For each test, the p-value of the test was equal to the proportion of samples in the null distribution greater than the test statistic. These 'exact p-values' were then corrected for the finite number of permutations performed (*Phipson and Smyth, 2010*). We chose not to analyze the effect of RT on the physiological signals as a simple scaling. Instead, we modeled the influence of RT on the fMRI signal via the weighted sum of three components, two of which are determined by RT (*de Gee et al., 2014*). This approach is expected to be more sensitive in revealing the influence of RT.

## Data cleaning

The first trial of each run was removed. This was done for all data types: fMRI, physiological, head movement, and behavioral data. Trials on which the observer did not respond were removed and analyzed separately as 'lapse trials.' No feedback tone was played on lapse trials. The lapse rate was 4.48% for hard runs and 6.68% for easy runs. We expected a larger lapse rate for easy runs because alertness was expected to be lower, which would cause more lapses.

## General linear mixed model

Our aim was to test the hypothesis that task difficulty and behavioral performance modulate the amplitude of TRRs in V1. We used a standard general linear model approach, with additional random intercepts and slopes for each observer to account for systematic differences in the fMRI signal across observers, i.e., a GLMM (*Chen et al., 2013*). We included all predictors of interest in the model: putative TRR components (trial onset ['fMRI_TO'], button press ['fMRI_BP'], and time on task ['fMRI_ ToT']) (*Yarkoni et al., 2009*), task difficulty, behavioral performance, predictions of the fMRI signal evoked by respiration and cardiac activity, and head movement. The GLMM had 24 fixed effects coefficients, corresponding to the following predictors: the convolution of a parametric HRF (the sum of two gamma functions) (*Shan et al., 2014*) fit separately to each observer's grand mean TRR with impulses at (1) trial onset and (2) button press, and with a (3) time-on-task boxcar between them, as well as (4) task difficulty, (5) behavioral accuracy, predictions for fMRI activity evoked by (6) respiration (*Birn et al., 2008*) and (7) cardiac activity (*Chang et al., 2009*), (8–13) the movement of the head in six dimensions (roll, pitch, yaw, y, x, and z) and (14–24) the interactions of all fMRI, task, and behavioral predictors (every combination). All predictors were downsampled (and appropriately low-pass filtered first, i.e., decimated) to the sampling rate of the fMRI signal. We also included one random intercept for each observer, and random slopes for the three fMRI predictors for each observer, for a total of 63 random effects coefficients. Each predictor was a time course the same length as the fMRI signal across all trials and runs, and for all observers. All predictors were included in a single design matrix, which was used to predict the trial-to-trial TRR in ipsilateral (right) V1. Head movement, cardiac, and respiration predictors were included to partial out their influence on the fMRI signal.

We used the fitlme function in Matlab to fit the GLMM to the voxel-averaged TRR (rV1) time course. We tested the assumptions of the linear model by examining the residuals as a function of the fitted values, and found that there was no trend, just a large Gaussian-looking cloud centered at zero. The distribution of residuals appeared Gaussian and the qq-plot showed slight deviation from Gaussian behavior at the tails, but Gaussian behavior in the center of the distribution. The total $R^2$ of the model was 4.5%, suggesting that the inclusion of the multiple interaction terms in the model didn't cause overfitting. The total $R^2$ of the model was low in comparison to linear models of stimulus-evoked fMRI activity, but nonetheless many of the regressors were statistically significant. Due to the paucity of studies that have modeled the TRR, it's unclear what one might expect this $R^2$ to be, given the intrinsic variability of the TRR and measurement noise. Next, we examined the full time course of the model prediction vs. the data, and observed that the random slopes we included for the fMRI predictors were necessary to capture sometimes large difference in the fMRI signal amplitude across observers. To compute p-values and $F$ statistics for linear combinations of the predictors (e.g. the combination of the three fMRI inputs, and its interaction with difficulty and/or accuracy), we performed an $F$-test using the coefTest function in Matlab.

To build the respiration and cardiac fMRI predictors, we convolved the raw signal from the respiration belt with a canonical respiration response function (computed at the same sampling rate), which effectively models the influence of respiration on the fMRI signal (*Birn et al., 2008*). We did the same with the cardiac (pulse oximeter) signal, based on ref (*Chang et al., 2009*). Only after computing the

convolutions at the intrinsic sampling rates of the respiration/cardiac measurements did we downsample these predictions to the sampling rate of the BOLD signal.

Note that we observed large and widespread hemodynamic responses in ipsilateral early visual cortex in the raw fMRI-BOLD signal and after every type of preprocessing procedure we tried (ME-ICA, global signal regression, and physiological artifact regression). We found that it was not possible to remove the TRR by 'denoising' the fMRI time series in preprocessing.

For each observer, we fit six parameters of a parametric HRF (the sum of two gamma functions) (*Shan et al., 2014*) to their grand mean trial-averaged TRR using fminsearch in Matlab. We could only use the 16 s of the grand mean TRR that we had to fit the parameters of the parametric form, but we allowed the resulting parametric HRF to be 32-s long. We used this best-fit parametric HRF to build the three fMRI predictors (fMRI_TO, fMRI_BP, and fMRI_ToT) by convolving the HRF with the impulses or boxcar occurring every trial (*Yarkoni et al., 2009*).

### Pupillometry

The experiment was repeated outside the scanner with an identical protocol, except with a shorter, 4-s ISI (*Burlingham et al., 2022*). Pupil area was recorded continuously during the task using an Eyelink 1000 infrared eye tracker (SR Research Ltd., Ontario, Canada) with a sampling rate of 500 Hz. Nine-point calibration and validation were performed before each run. Blinks and saccades were removed from the pupil size time series and interpolated over (using piecewise cubic interpolation). Pupil size time series from each run were band-pass filtered with a fourth order zero-phase (i.e. 'filtfilt' in Matlab) Butterworth filter with 0.03 Hz and 10 Hz cutoffs (*Burlingham et al., 2022*). Average pupil responses, time-locked to trial onset, were calculated for hard and easy runs, correct and incorrect trials separately. The SEM for each average time series was computed across all runs and observers.

### Supporting information

### Global signal does not show same dependence on task difficulty and behavioral performance that the TRR does

We repeated the same GLMM that we performed on the TRR, but instead tested the global signal to see whether it shows a similar dependence on task difficulty and behavioral performance. There was only one statistically significant predictor out of nine, which was the interaction of task difficulty and the fMRI_BP predictor (p=0.02). Additionally, the *F* statistic (5.04) was smaller for the global signal than it was for the TRR in V1, for this interaction. Correspondingly, the coefficient test was only significant for difficulty, not accuracy, nor the interaction of difficulty × accuracy. These observations suggest that the global signal may depend on task difficulty, but to a smaller degree than the TRR. This is not surprising, given that the global signal covaries with artifactual physiological influences on the BOLD signal (such as head motion), and these artifacts are themselves known to depend on task difficulty. Overall, we conclude that the global signal shows a weaker and different dependence on task difficulty and behavioral performance than the TRR in early visual cortex.

### TRRs in V1, V2, and V3 largely showed statistically significant relationships with task difficulty and behavioral performance

We found that TRR amplitudes were largest in V1, and progressively smaller in V2 and V3. To more closely examine the statistical relationships between TRR amplitude, task difficulty, and behavioral performance, we conducted the same GLMM in V2 and V3 that we conducted in V1. *F* values for the coefficient tests were as follows — difficulty (V1/V2/V3): 6.61, 3.05, 7.96; accuracy (V1/V2/V3): 3.81, 0.48, 3.43; difficulty × accuracy (V1/V2/V3): 3.79, 0.91, 3.78. All p-values were statistically significant except for accuracy and difficulty × accuracy in V2. Overall, this shows task difficulty modulated TRR amplitude within V1, V2, and V3, and that accuracy significantly modulated TRR amplitude in V1 and V3, but to a lesser degree in V2.

### Lack of support for the hypothesis that contralateral connections in visual cortex cause the TRR

Previous studies have hypothesized that contralateral anatomical connections between the two hemispheres of visual cortex lead to coordinated retinotopically organized activity in left and right hemisphere visual cortex (*Stark et al., 2008*; *Arcaro et al., 2015*). If so, one might imagine that our visual

stimulus, which was presented in the right hemifield only, may have evoked a neural response in the right hemisphere as well as the left that could explain away our results. To test this idea, we took voxels from the right hemisphere representing only the upper visual field where we could be confident that there was no visual response (because our stimulus was always in the lower visual field) and reran the main GLMM analysis, which tests for modulations of the TRR with difficulty and behavioral performance. This analysis returned the same statistical outcomes as when we initially included both the upper and lower visual fields, with p-values of $1.85 \times 10^{-4}$, 0.0096, and 0.0099 ($F$: 6.61, 3.81, 3.79) for the effects of difficulty, accuracy, and the interaction difficulty × accuracy, respectively. We repeated this upper-visual-field-only analysis on V2 and V3 and still found statistically significant effects for all comparisons (except the effect of accuracy in V2, for which $p>0.05$). These results are inconsistent with the hypothesis that the TRR arises from stimulus-evoked activity showing up in the ipsilateral hemisphere via interhemispheric connectivity.

## Stimulus-evoked activity during the localizer

Each observer completed a stimulus localizer scan, in which a stimulus was presented in the location of the stimulus during the task (right hemifield) or in its mirror location in the left hemifield. The stimulus was a grating identical to that used in the main experiment, except that it remained on the screen continuously for 9 s and changed phase and orientation every 200 ms to prevent adaptation (*Figure 5—figure supplement 1*). To analyze these data, we performed a correlation analysis in which we identified fluctuations in the fMRI signal matching the temporal frequency with which the stimulus alternated visual field (left-right change every 18 s). We observed robust stimulus-evoked activity in antiphase between the two hemispheres, as expected (*Figure 5—figure supplement 1*). That is, the fMRI signal peaked in the left hemisphere at a fixed delay after the stimulus was presented in the right hemifield, and the fMRI signal peaked in the right hemisphere the same delay after the stimulus was presented in the left hemifield. We projected this stimulus-evoked activity into the space of the visual field using a pRF analysis first introduced in ref (*Roth et al., 2020*) (see their methods for details) (*Figure 5—figure supplement 1*). This analysis produced two foci of activity nearly antiphase with each other in the lower left and right hemifield at the location of the two stimuli in the localizer experiment.

The stimulus-evoked responses we observed (in brain space; *Figure 5—figure supplement 1*) during the localizer were much more circumscribed compared to the responses observed during the task in the left hemisphere, which were a combination of stimulus-evoked, attentional, and task-related responses (*Figure 5—figure supplement 2*), and compared to the responses in right hemisphere, which were just task-related (*Figure 2*). The amplitude of the fMRI response in the right hemisphere (measured as std of time course) was similar during the localizer and task (0.30% vs. 0.28% change, median across all voxels in right V1, all runs, and all observers). And likewise for the left hemisphere (localizer vs. task, 0.32% vs. 0.30% change). That said, it is unclear how stimulus-evoked, attentional, and task-related responses interact in visual cortex. This is an important question, and one that has been addressed extensively in the attention literature, where attention has been shown to have multiplicative effect on stimulus-evoked activity (*Herrmann et al., 2010*; *Carandini and Heeger, 2011*). We couldn't investigate this sort of question in our data because we purposely designed our task to minimize the stimulus-evoked activity (note that stimulus duration was 9 s in the localizer vs. only 200 ms in the task, and the stimulus was only 1.5° in size). For this same reason, a detailed comparison of stimulus- and task-driven activity is beyond the scope of this paper. However, see refs (*Cardoso et al., 2012*; *Cardoso et al., 2019*; *Herman et al., 2017*) for some first hints about these questions. *Roth et al., 2020* (*Roth et al., 2020*), in which the authors excluded voxels responsive to the stimulus in the localizer and still observed robust TRRs that were significantly amplitude modulated by reward magnitude.

## Correction for multiple comparisons

A total of 128 hypothesis tests were performed throughout the paper. We note that not all tests were independent, but that positive dependencies were expected to exist between them because they were computed using the same test statistics. To control the overall type I error rate at an α level of 0.05 while assuming dependencies between tests, we subjected all exact p-values to a 'two-stage' adaptive procedure (*Benjamini et al., 2006*; *Groppe et al., 2011*). It has been shown via simulation that of a number of similar tests, this adaptive test is best at controlling false discovery rate, regardless of effect size, in the presence of positive dependence (*Stevens et al., 2017*). This procedure yielded a corrected critical p-value, α = 0.0491,

which was adopted as a conservative cutoff to judge the strength of the statistical evidence for each hypothesis test. We found no p-values between 0.0491 and 0.05, so we treated our effective α as 0.05 throughout the text.

## Acknowledgements

Thanks to Aniruddha Das and Jonathan Winawer for their comments on the manuscript and Tobias Donner for helpful discussion. This research was supported by National Eye Institute grant R01-EY025330 (to D.J.H.), the National Eye Institute Visual Neuroscience Training Grant, T32-EY007136 (to C.S.B. through NYU), and the National Defense Science and Engineering Graduate Fellowship (to C.S.B.), and by the Intramural Research Program of the National Institutes of Health (ZIA-MH-002966) - National Institute of Mental Health Clinical Study Protocol 93 M-0170, NCT00001360.

## Additional information

### Funding

| Funder | Grant reference number | Author |
|---|---|---|
| National Eye Institute | R01-EY025330 | David J Heeger |
| National Institute of Mental Health | ZIA MH002966 | Elisha P Merriam |
| National Eye Institute | T32-EY007136 | Charlie S Burlingham |
| National Defense Science and Engineering Graduate Fellowship | | Charlie S Burlingham |

The funders had no role in study design, data collection and interpretation, or the decision to submit the work for publication.

### Author contributions

Charlie S Burlingham, Formal analysis, Investigation, Methodology, Software, Visualization, Writing – original draft, Writing – review and editing; Minyoung Ryoo, Investigation, Writing – review and editing; Zvi N Roth, Investigation, Methodology, Writing – review and editing; Saghar Mirbagheri, Formal analysis, Investigation, Methodology, Writing – review and editing; David J Heeger, Conceptualization, Funding acquisition, Supervision, Writing – review and editing; Elisha P Merriam, Conceptualization, Funding acquisition, Investigation, Project administration, Resources, Software, Supervision, Writing – review and editing

### Author ORCIDs

Charlie S Burlingham http://orcid.org/0000-0002-9286-2091
Zvi N Roth http://orcid.org/0000-0002-2173-1625
Saghar Mirbagheri http://orcid.org/0000-0002-0994-3932
David J Heeger http://orcid.org/0000-0002-3282-9898
Elisha P Merriam http://orcid.org/0000-0003-2787-566X

### Ethics

Human subjects: Informed consent, and consent to publish, was obtained from all human participants. The consent and experimental protocol were in compliance with the safety guidelines for MRI research, and were approved by both the University Committee on Activities involving Human Subjects at New York University (RB-FY2016-158), and the Institutional Review Board at the National Institutes of Health (93-M-0170, ClinicalTrials.gov identifier: NCT00001360).

### Decision letter and Author response

Decision letter https://doi.org/10.7554/eLife.73018.sa1
Author response https://doi.org/10.7554/eLife.73018.sa2

## Additional files

### Supplementary files
• Transparent reporting form

• Source code 1. GLMM coefficients and p-values. See attached fileS1.txt file for full output from Matlab's lme function.

### Data availability
All data and code are available at osf.io/8fp35 (https://doi.org/10.17605/OSF.IO/8FP35).

The following dataset was generated:

| Author(s) | Year | Dataset title | Dataset URL | Database and Identifier |
|---|---|---|---|---|
| Burlingham CS, Ryoo M, Roth ZN, Mirbagheri S, Heeger DJ, Merriam EP | 2022 | Task-related hemodynamic responses in human early visual cortex are modulated by task difficulty and behavioral performance | http://osf.io/8fp35 | Open Science Framework, 10.17605/OSF.IO/8FP35 |

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
