## [Editor Report]

This work is of general interest to those using hemodynamic imaging, such as fMRI, to study the brain. A hemodynamic signature that is modulated by arousal level changes on a trial-to-trial basis, such as those evoked by a difficult task, both provides insight into arousal influences on cortical activity and characterize a prominent signal in hemodynamic data that is rarely considered.

---

## [Decision Letter]

**Decision letter after peer review:**

Thank you for submitting your article "Task-related hemodynamic responses in human early visual cortex are modulated by task difficulty and behavioral performance" for consideration by *eLife*. Your article has been reviewed by 3 peer reviewers, one of whom is a member of our Board of Reviewing Editors, and the evaluation has been overseen by a Reviewing Editor and Floris de Lange as the Senior Editor. The following individual involved in review of your submission has agreed to reveal their identity: Kristina Visscher (Reviewer #2).

Essential revisions:

Burlingham and colleagues studied a number of individuals using fMRI while they performed an orientation discrimination task. In this task they were able to isolate a task-related response (TRR) in the fMRI BOLD signal, that was present in visual cortex in the hemisphere ipsilateral to the stimulus, independent of other physiological variables such as respiration, and dependent on the difficulty of the task. They use a mathematical model to isolate the magnitude of the TRR on individual trials, and link it to shifts in arousal.

The reviewers agreed that this work was clearly interesting to the field of fMRI researchers, and extends previous research by the authors and others showing what is contained in the TRR. In order to make this work suitable for publication in *eLife* and extend the reach of the work to those beyond this specialized field, the reviewers agreed that the authors would have to demonstrate two key points about the TRR:

1. Demonstrate convincingly that the TRR shown here, and its properties, are a new feature of the brain's activity that is being described and uncovered – i.e., that it could not simply be an artifact, either of signal due to retinal responses to the stimuli (due to eye movements in the scanner), connections to contralateral V1 which we know receives inputs from the stimuli, respiration, and maybe other things.

2. Link the TRR and the work here convincingly to other literature and work outside the field of those studying TRR, to provide a computational or mechanistic interpretation of this phenomenon. For example, there are numerous studies of the pupil and arousal, neurovascular coupling and arousal, and broadly how arousal modulates neurons and behavior. Links between the TRRs observed here and that work would make its broad appeal clearer.

Below is a more detailed description of the issues related to concerns 1 and 2 above, as well as a compiled list of specific additional comments from all reviewers.

(1A) The authors state that because the measurements were collected from ipsilateral V1 and "based on numerous prior studies," the TRR was unlikely to be a stimulus or attention evoked response (lines 70-73). Do the authors have any data from the contralateral hemisphere to demonstrate the difference between the evoked and task-related response or can they elaborate more on the prior work that distinguishes these responses? This would be helpful given the first major concern about the subject's eye position during the task. Also, spatial attention modulates activity both at the site of the attention and elsewhere in single-neuron recordings. Can the authors elaborate a bit on what these prior studies show that rules out attention here? It seems like showing the comparison of contralateral and ipsilateral responses in this case, at least as a supplement, would be useful.

It would be informative to know what the patterns of activity in contralateral visual cortex are. If it is the case that the same patterns exist in the contralateral cortex (e.g. effects of 'arousal' are similar), then a possible interpretation would be that the strong connections between homologous brain areas drives the effects seen in the ipsilateral cortex. However, if the patterns in the contralateral cortex do not show a hint of the effect shown in ipsilateral cortex (e.g. the effects shown in ipsilateral cortex are still present after regressing out effects in contralateral cortex, or the effects in contralateral cortex were very close to 0) this would strengthen the claim that the effects observed are independent of contralateral cortex.

B) During the main experimental paradigm in the MRI scanner, the subject was instructed to centrally fixate throughout the session. Was there any eye tracking to ensure they maintained fixation within the scanner or to remove trials where they looked towards the target? Otherwise, the subject could have developed a strategy, in the extreme, to look towards or directly at the target during more difficult trials (or something more subtle but equally problematic). Eye movements, both small saccades or larger ones that shift the visual field, could account for some aspect of the task-related hemodynamic response. It's also unclear how, if at all, the authors handled this concern in their second experiment with pupillometry repeated outside the scanner, where they clearly do have the eye tracking data. One way to test for this effect in data with pupillometry would be to compare the variance in eye stability (that is, how well were they fixating) between conditions (easy vs. hard, correct vs. incorrect, slow RT vs. fast RT). Although they may not have the eye tracking data in the scanner, at least in the analogous experiment outside they should better analyze the eye data to determine whether the fixation was similar in the different conditions. Changes in fixation can also impact pupillometry data as well, another reason to do this control analysis.

Overall, the lack of fixation tracking poses a concern to the claim that TRRs do not arise from spatial attention (who knows if people really fixated). While acknowledging that rectifying this would only be possible with new data, the authors would be encouraged to collect eye-tracking data in the fMRI in the future. There may even be publicly available datasets of fMRI + eye tracking, allowing them to account for potential fixation confounds in follow-up work.

(2A) The authors claim the results in Figure 4 suggest arousal as a common driver for the modulation of the TRR by task difficulty and performance. However, this statement seems weakly supported given most of the physiologic metrics don't show significant effects across conditions. While pupil changes appear consistent with the fMRI results, concerns about the subject failing to maintain fixation in certain conditions could potentially contribute to these results. Furthermore, the effect of trial difficulty on trial-averaged pupil dynamics is not a new result (see Urai et al. 2017, Nat. Comm) and the authors should place their findings in context of those results, which demonstrated similar trends.

B) The paper does not seem to consider the potential computational nature of TRRs (figure 3) and the other physiological measures (figure 4). Beyond quantifying their dependence on difficulty, RT and lapses, would there be a unified model that could explain all of these (e.g. confidence or prediction errors, which are known to predict pupil responses? https://www.nature.com/articles/s41598-018-31985-3). If the authors could link their finding to the literature on decision-making and psychophysiology, the relevance of the work would extend much beyond the fMRI community.

C) Related to these points, the introduction is very brief and centered entirely on the phenomenon of the TRR, without placing it very firmly in the broader context of the literature of arousal, decision-making, etc, to extend the relevance of the work beyond the fMRI community. Similarly, the discussion presents potential vascular origins of the TRR, but only very briefly as an aside mentions neuromodulation, subthreshold activity, and subpopulations of neurons as possible explanations. Many of these have been studied extensively in other contexts in linking fMRI results to neuronal activity, and could be discussed here in better linking this work to the broader field.

Additional specific comments:

– I find the color coding in figure 4 very confusing: I'm used to correct = green and error = red. Perhaps changing this would make the figure easier to parse.

– what is the sampling rate of the pulse oximeter? If this signal (as well as respiration) is significantly down-sampled with respect to the fMRI TR, could this lead to an under correction of physiological noise?

– How many trials were lapses? Did this vary across the session, or with hard/easy blocks?

– It is difficult to judge whether the pre-processing steps (physiological noise removal, global signal regression) are appropriate and sufficient. However, it's likely that the measurement of TRRs strongly depends on these steps – for the non-fMRI expert, it would help to read a short statement describing the choice of procedures.

– The analyses done to compare TRRs in V2 and V3 were unclear- was another GLMM constructed or are the authors simply comparing the trial-averaged responses across conditions? Secondly, no statistics are shown. If they want to claim smaller effects across the visual hierarchy, they should test that specifically (i.e., test whether V2's effects are < V1, or V3 < V2 < V1). Were the difficulty and performance related effects on TRRs in V2 and V3 significant? Or just visually smaller than V1? Lastly, the shape of the TRRs in V2 and V3 seem more similar to the lapsed trials in Figure 3e. Some discussion of the shape of the TRRs might be warranted (especially given lapse trials in 3e and V2 and V3 in 5c seem to have an "initial dip" in the BOLD signal and deviate from the typical TRR shape).

– Lines 129-131 (RT analysis)- did the proportion of trials with fast vs. slow RTs vary with task difficulty or performance? For example, did the subjects tend to respond faster overall in the more difficult trials? Also, it would be helpful if the prediction for a relationship between RT and arousal was more specific or placed in context of RT changes with arousal in the literature (ex. elevated arousal is associated with faster RTs).

– Lines 133-136 ". first, by the latency between the trial onset and button press inputs, and second, by the duration of the time-on-task input (Figure S1). We hypothesized that amplitude modulation of the TRR with reaction time (on individual trials) was determined by the sum of these three hypothesized inputs." Those are two hypothesized inputs. What is the third?

– Why refer to "time on task" as wholly separate from Reaction Time – they are the same construct. I think it would be easier to follow your argument if you used the same word for them. If there is a separate construct I'm missing, then that needs to be clearer. On line 133, you say that latency was represented by latency between trial onset and button press, and second by duration of the time on task input. This needs to be explained better, because from what I know at this point in the paper, time on task IS the time difference between trial onset and button press. I understand why you would want the three components in your model, but not why you would refer to the effects of a fMRI_TO+fMRI_BP as a "reaction time" measurement. Perhaps you are using the term in a special way? (do you mean RT as "time of responses" rather than the more common "time between stimulus and response?")

– Also, in for example Figure 5, you don't include Easy incorrect trials (because there aren't enough trials to compute temporal variability), but you can compute e.g. % change for those trials. I don't see why the power should make it impossible to calculate the Easy I condition variability (Figure 5D), but give reliable estimates of the mean and SD for the Easy I condition. A quick power analysis here would be sufficient.

– Was there a tone presented non lapse trials? If so, the logic of the lapse trials amplitude being bigger than response trials could be the same as the easy error trials having stronger responses than the correct trials: the participant is surprised and brought back to full attention to the task.

– Line 139- add some indication of how the simulation was performed (presumably using the GLMM?)

– It isn't clear why reaction time was not also addressed in the analysis of physiological data, as it was in the hemodynamic data. Some of the bigger effects in Figure 3 were for fast vs. slow RTs.

– Figure 3c-d- it's initially confusing that the color scheme for fast/slow RT is the same as the correct/incorrect trials in 3b

– Figure 3f-g- why are there no statistics demonstrating the significance of these results?

– Figure 3G legend: "viability" should be "variability."

– Line 167- an additional line providing details on how the circular standard deviation of the Fourier phase serves as a measure of temporal variability would provide clarity.

– Line 168- the authors imply that temporal precision was highest when arousal level was higher. Some brief discussion of this point in the Discussion section would be interesting.

– Line 177 (pupil analysis)- although potentially outside the scope of this paper, it might be valuable to also assess microsaccade rate, which has been linked to arousal, in different task/performance conditions

– Line 189- it should be clearer that all the other physiologic metrics besides pupil were measured in the scanner simultaneously during the fMRI experiment

– Figure 4: For each physiologic metric, the authors are making multiple comparisons across task/performance conditions but don't discuss any statistical correction for multiple comparisons. Do the significant trends in this figure survive that correction?

– Figure 4: Was each metric (ex. heart rate, pupil) averaged over the whole trial or some smaller bin within the trial? This detail should be clearer in the methods. If the whole trial was used, could some of the pupil effect be related to the pupillary light reflex evoked by the stimulus flash? Alternatively, if a smaller bin near the feedback tone was used, this could provide a better estimate of the surprise-related arousal effect the authors mention.

– Line 290 -> pupil dilation has been found to scale with many neuromodulatory (and other cortical and subcortical) systems (e.g. de Gee et al. 2017, Cazettes et al. 2021, Joshi); I find it too strong to equate pupil fluctuations with the LC-NE system.

– Global signal- if some aspect of the task-related response was present across cortex (consistent with a global arousal signature), then it might be present in the global signal, which was regressed out in this work because it also contains elements of physiologic processes. Potentially outside the scope of this work, but does this trial-to-trial analysis method find a similar TRR in the global signal?

– Prior work on TRRs demonstrated "ramp-like" increases and decreases in hemodynamics over minutes between blocks of high and low reward conditions (Cardoso et al. 2019, PLOS). It would be interesting, although potentially outside the scope of this work, if the authors saw similar trends between easy and difficult task blocks or could model this effect in their GLMM by adding trial number within the block as a variable.

– As the authors know, BOLD fMRI studies show robust attention effects in V1 that are much less evident in single-unit studies. This raises the point of how the effects of attention on individual neurons, and in brain areas, relate to the pooled fMRI signal. A similar point could be made here. Perhaps it's not that the effects are smaller across the hierarchy, but rather that the way they manifest across the hierarchy makes them less visible to the MRI signal? Certainly at least one neuromodulator that is thought to relate to arousal, norepinephrine, seems to be more strongly distributed to PFC than to sensory cortical areas (Foote and Morrison, Annual Review of Neurosci 1987, among many suitable papers to reference). I would suggest some discussion of this literature, and idea, as a caveat to this conclusion if it remains.

---

## [Author Response]

Essential revisions:Burlingham and colleagues studied a number of individuals using fMRI while they performed an orientation discrimination task. In this task they were able to isolate a task-related response (TRR) in the fMRI BOLD signal, that was present in visual cortex in the hemisphere ipsilateral to the stimulus, independent of other physiological variables such as respiration, and dependent on the difficulty of the task. They use a mathematical model to isolate the magnitude of the TRR on individual trials, and link it to shifts in arousal.The reviewers agreed that this work was clearly interesting to the field of fMRI researchers, and extends previous research by the authors and others showing what is contained in the TRR. In order to make this work suitable for publication in eLife and extend the reach of the work to those beyond this specialized field, the reviewers agreed that the authors would have to demonstrate two key points about the TRR:1. Demonstrate convincingly that the TRR shown here, and its properties, are a new feature of the brain's activity that is being described and uncovered – i.e., that it could not simply be an artifact, either of signal due to retinal responses to the stimuli (due to eye movements in the scanner), connections to contralateral V1 which we know receives inputs from the stimuli, respiration, and maybe other things.

We have addressed each of these alternative explanations for TRRs and we are confident that the TRR is not a trivial artifact of any of these factors. Regarding respiration, we have addressed this on lines 59-64 and 81-87. See below for detailed replies about each of the other alternatives.

2. Link the TRR and the work here convincingly to other literature and work outside the field of those studying TRR, to provide a computational or mechanistic interpretation of this phenomenon. For example, there are numerous studies of the pupil and arousal, neurovascular coupling and arousal, and broadly how arousal modulates neurons and behavior. Links between the TRRs observed here and that work would make its broad appeal clearer.

We have substantially revised the discussion to link our findings with the broader literature, specifically focusing on possible computational and mechanistic interpretations of TRRs. Please see the new sections of the discussion starting on lines 364, 424, and 447.

Below is a more detailed description of the issues related to concerns 1 and 2 above, as well as a compiled list of specific additional comments from all reviewers.(1A) The authors state that because the measurements were collected from ipsilateral V1 and "based on numerous prior studies," the TRR was unlikely to be a stimulus or attention evoked response (lines 70-73). Do the authors have any data from the contralateral hemisphere to demonstrate the difference between the evoked and task-related response or can they elaborate more on the prior work that distinguishes these responses? This would be helpful given the first major concern about the subject's eye position during the task.

In each scanning session, in addition to the main experiment we ran a stimulus localizer that allowed us to identify the region of cortex responding to the stimulus, as well as to the mirror-symmetric location in the opposite hemifield. This localizer also allowed us to analyze portions of visual cortex that were not responsive to the visual stimulus. In the revision, we now report these results in the supplement section “Stimulus-evoked activity during the localizer” and the new SI Figure S4. The spatial profile of the stimulus-evoked activity was very different from the spatial profile of the TRR. Specifically, stimulus-evoked activity was concentrated at the retinotopic location of the stimulus, as expected, and the TRR was widespread throughout early visual cortex, even in the absence of a stimulus. Please also see Roth et al. 2020 (*PLOS Biology*), a related study that used the same stimulus and a similar task protocol, for additional analyses that address this same question. The issue of whether stimulus-evoked activity and TRRs interact is outside the scope of this work, and one would require an entirely different experimental protocol to properly address (e.g., see Cardoso et al., *Nat. Neurosci.,* 2012 in which this topic is studied in monkey V1 using intrinsic signal optical imaging).

Also, spatial attention modulates activity both at the site of the attention and elsewhere in single-neuron recordings. Can the authors elaborate a bit on what these prior studies show that rules out attention here? It seems like showing the comparison of contralateral and ipsilateral responses in this case, at least as a supplement, would be useful.

We have now discussed this literature (see lines 387-398). For the visual comparison you suggested, please see Figure S5.

It would be informative to know what the patterns of activity in contralateral visual cortex are. If it is the case that the same patterns exist in the contralateral cortex (e.g. effects of 'arousal' are similar), then a possible interpretation would be that the strong connections between homologous brain areas drives the effects seen in the ipsilateral cortex. However, if the patterns in the contralateral cortex do not show a hint of the effect shown in ipsilateral cortex (e.g. the effects shown in ipsilateral cortex are still present after regressing out effects in contralateral cortex, or the effects in contralateral cortex were very close to 0) this would strengthen the claim that the effects observed are independent of contralateral cortex.

Please see the new sections of the supplement “Lack of support for the hypothesis that contralateral connections in visual cortex cause the TRR” and “Stimulus-evoked activity during the localizer” as well as the new Figures S4 and S5, which address these concerns.

B) During the main experimental paradigm in the MRI scanner, the subject was instructed to centrally fixate throughout the session. Was there any eye tracking to ensure they maintained fixation within the scanner or to remove trials where they looked towards the target? Otherwise, the subject could have developed a strategy, in the extreme, to look towards or directly at the target during more difficult trials (or something more subtle but equally problematic). Eye movements, both small saccades or larger ones that shift the visual field, could account for some aspect of the task-related hemodynamic response. It's also unclear how, if at all, the authors handled this concern in their second experiment with pupillometry repeated outside the scanner, where they clearly do have the eye tracking data. One way to test for this effect in data with pupillometry would be to compare the variance in eye stability (that is, how well were they fixating) between conditions (easy vs. hard, correct vs. incorrect, slow RT vs. fast RT). Although they may not have the eye tracking data in the scanner, at least in the analogous experiment outside they should better analyze the eye data to determine whether the fixation was similar in the different conditions. Changes in fixation can also impact pupillometry data as well, another reason to do this control analysis.

We have eye data from the pupillometry recordings outside the scanner. We have reanalyzed these data, comparing the variance in eye position between conditions. We did not find any statistically significant differences in this measure of fixation stability between conditions. Please see this analysis on lines 251-268.

If the widespread BOLD responses were luminance-evoked, i.e., caused by eye movements changing the pattern of retinal stimulation, we would have expected to observe the strongest responses in V1 at eccentricities close to the representation of the screen edge (where visual contrast was highest). To test this hypothesis, we analyzed the BOLD signal amplitude across the representation of the visual field, but did not observe larger response amplitudes in the cortical locations corresponding to the edge of the screen. See lines 270-284.

Overall, the lack of fixation tracking poses a concern to the claim that TRRs do not arise from spatial attention (who knows if people really fixated). While acknowledging that rectifying this would only be possible with new data, the authors would be encouraged to collect eye-tracking data in the fMRI in the future. There may even be publicly available datasets of fMRI + eye tracking, allowing them to account for potential fixation confounds in follow-up work.

Thank you for the suggestion and we plan to collect eye tracking data in the scanner going forward. We found that the same observers fixated at the cross faithfully outside the scanner during the pupillometry experiment. Together with the screen-edge-eccentricity analysis, this implies that eye movements do not underlie the TRR.

(2A) The authors claim the results in Figure 4 suggest arousal as a common driver for the modulation of the TRR by task difficulty and performance. However, this statement seems weakly supported given most of the physiologic metrics don't show significant effects across conditions. While pupil changes appear consistent with the fMRI results, concerns about the subject failing to maintain fixation in certain conditions could potentially contribute to these results.

Most of the physiological metrics (6/7) show significant modulations with either difficulty or accuracy. Given that they each reflect different aspects of autonomic nervous system activity, we didn’t expect that they would all be modulated in the exact same way as each other or the TRR (discussed on lines 581-598).

Pupil size is the least noisy metric and shows the strongest effects. Many other papers show similar modulations of pupil size with difficulty and accuracy while also controlling for eye movements, so we are confident our results are not artifactual (e.g., Kahneman and Beatty 1966, 1967; Urai et al. 2017; Burlingham, Mirbagheri, and Heeger 2022). See lines 210-211.

Furthermore, the effect of trial difficulty on trial-averaged pupil dynamics is not a new result (see Urai et al. 2017, Nat. Comm) and the authors should place their findings in context of those results, which demonstrated similar trends.

Agreed. See lines 210-211.

B) The paper does not seem to consider the potential computational nature of TRRs (figure 3) and the other physiological measures (figure 4). Beyond quantifying their dependence on difficulty, RT and lapses, would there be a unified model that could explain all of these (e.g. confidence or prediction errors, which are known to predict pupil responses? https://www.nature.com/articles/s41598-018-31985-3). If the authors could link their finding to the literature on decision-making and psychophysiology, the relevance of the work would extend much beyond the fMRI community.

We have added a discussion about the possible computational nature of TRRs, and linked our findings to the study you mentioned as well as De Gee et al. 2017, and other relevant papers. See the new paragraphs of the Discussion starting on lines 364, 424, and 447. We have also modified the introduction on lines 41-45 in order to highlight this possibility. Thank you for the suggestion.

C) Related to these points, the introduction is very brief and centered entirely on the phenomenon of the TRR, without placing it very firmly in the broader context of the literature of arousal, decision-making, etc, to extend the relevance of the work beyond the fMRI community. Similarly, the discussion presents potential vascular origins of the TRR, but only very briefly as an aside mentions neuromodulation, subthreshold activity, and subpopulations of neurons as possible explanations. Many of these have been studied extensively in other contexts in linking fMRI results to neuronal activity, and could be discussed here in better linking this work to the broader field.

We have modified the introduction to better connect our research to the relevant literatures on arousal and decision-making. And we have expanded our discussion of the origins of the TRR, specifically focusing on possible neural signals that could give rise to TRRs and the associated empirical support for these hypotheses. See the new paragraphs of the Discussion starting on lines 364, 424, and 447.

Additional specific comments:– I find the color coding in figure 4 very confusing: I'm used to correct = green and error = red. Perhaps changing this would make the figure easier to parse.

We understand the reviewer’s color preference. We have already published other papers with this particular color scheme (Burlingham, Mirbagheri, Heeger, 2022), where difficulty is red/green, and accuracy is light/dark colors. So we prefer to leave the color scheme the same for consistency across publications.

– What is the sampling rate of the pulse oximeter? If this signal (as well as respiration) is significantly down-sampled with respect to the fMRI TR, could this lead to an under correction of physiological noise?

The sampling rate of the pulse oximeter was 50 Hz, now reported on lines 570-573.

All physiological measurements were kept at the highest sampling rate possible (i.e., their original sampling rate) for all analyses. For estimating the physiological signatures plotted in Figure 4, we used the signals at this intrinsic sampling rate. For removing physiological artifacts from the TRR, we took the raw pulse oximeter and respiration belt signals and them convolved with an HRF (either cardiac HRF or respiratory HRF) that was at the same sampling rate, and only then down sampled the predicted physiologically-driven BOLD activity to the same sampling rate of the BOLD signal (0.67 Hz). Lowpass filtering with the HRF before down sampling avoids aliasing. So, the under correction scenario that the reviewer suggests is not an issue. These details are now reported on lines 668-673 of the methods.

– How many trials were lapses? Did this vary across the session, or with hard/easy blocks?

5.58% of all trials were lapses. Most runs (>90%) had zero lapsed trials, so there was variation across a session, but the amount of data is too low to detect anything systematic. For hard runs, the lapse rate was 4.48% (39/870 trials), and for easy runs, 6.68% (59/885 trials). We expected a larger lapse rate for the easy runs, because alertness was expected to be lower. We’ve included this information on lines 623-628.

– It is difficult to judge whether the pre-processing steps (physiological noise removal, global signal regression) are appropriate and sufficient. However, it's likely that the measurement of TRRs strongly depends on these steps – for the non-fMRI expert, it would help to read a short statement describing the choice of procedures.

Please see lines 59-64 for a description of these choices. We also added a statement about the impact of the specific choices of pre-processing steps on lines 675-678.

– The analyses done to compare TRRs in V2 and V3 were unclear- was another GLMM constructed or are the authors simply comparing the trial-averaged responses across conditions? Secondly, no statistics are shown. If they want to claim smaller effects across the visual hierarchy, they should test that specifically (i.e., test whether V2's effects are < V1, or V3 < V2 < V1).

We performed a permutation test to test for differences in the amplitude of the TRR across V1/2/3. This is now reported on lines 241-245.

Were the difficulty and performance related effects on TRRs in V2 and V3 significant? Or just visually smaller than V1?

We have now performed a separate GLMM for each visual area V1/2/3 with difficulty, accuracy, and their interaction as regressors. The effects of difficulty and accuracy were significant in V1, but not in V2 nor V3. This is now reported in the supplement in the section “TRRs in V1, V2, and V3 largely showed statistically significant relationships with task difficulty and behavioral performance.”

Lastly, the shape of the TRRs in V2 and V3 seem more similar to the lapsed trials in Figure 3e. Some discussion of the shape of the TRRs might be warranted (especially given lapse trials in 3e and V2 and V3 in 5c seem to have an "initial dip" in the BOLD signal and deviate from the typical TRR shape).

Interesting observation. We now discuss the shape differences across visual areas on lines 246-249.

– Lines 129-131 (RT analysis)- did the proportion of trials with fast vs. slow RTs vary with task difficulty or performance? For example, did the subjects tend to respond faster overall in the more difficult trials?

For Figure 3 we used a median split, so the proportion of “fast”/“slow” trials was always 50/50%, regardless of the trial type (e.g., easy-correct). We’ve clarified this on lines 133-134. To answer your second question, we ran a GLMM predicting RT from difficulty, accuracy, and their interaction. The results are now reported on lines 137-140. We also added a Figure 3D (inset), showing the average RTs across observers.

Also, it would be helpful if the prediction for a relationship between RT and arousal was more specific or placed in context of RT changes with arousal in the literature (ex. elevated arousal is associated with faster RTs).

We have specified a prediction for the relationship between arousal and RT, based on the literature, on lines 364-385.

– Lines 133-136 ". first, by the latency between the trial onset and button press inputs, and second, by the duration of the time-on-task input (Figure S1). We hypothesized that amplitude modulation of the TRR with reaction time (on individual trials) was determined by the sum of these three hypothesized inputs." Those are two hypothesized inputs. What is the third?

This was a typo. The three inputs are (1) a δ function at the trial onset time, (2) a δ function at the button press time, and (3) a boxcar extending between the trial onset and button press times. We have fixed this on lines 142-145.

– Why refer to "time on task" as wholly separate from Reaction Time – they are the same construct. I think it would be easier to follow your argument if you used the same word for them. If there is a separate construct I'm missing, then that needs to be clearer. On line 133, you say that latency was represented by latency between trial onset and button press, and second by duration of the time on task input. This needs to be explained better, because from what I know at this point in the paper, time on task IS the time difference between trial onset and button press. I understand why you would want the three components in your model, but not why you would refer to the effects of a fMRI_TO+fMRI_BP as a "reaction time" measurement. Perhaps you are using the term in a special way? (do you mean RT as "time of responses" rather than the more common "time between stimulus and response?")

See response to the preceding comment. We have clarified our ideas about these components further on lines 157-162.

– Also, in for example Figure 5, you don't include Easy incorrect trials (because there aren't enough trials to compute temporal variability), but you can compute e.g. % change for those trials. I don't see why the power should make it impossible to calculate the Easy I condition variability (Figure 5D), but give reliable estimates of the mean and SD for the Easy I condition. A quick power analysis here would be sufficient.

We’ve added it. We forgot to include this in the Figure 5 caption (as had already been done for Figure 3). See the new caption for the mean and SEM of the temporal variability.

– Was there a tone presented non lapse trials? If so, the logic of the lapse trials amplitude being bigger than response trials could be the same as the easy error trials having stronger responses than the correct trials: the participant is surprised and brought back to full attention to the task.

No tone was played on lapse trials. We now describe this on lines 623-628 of the Methods.

– Line 139- add some indication of how the simulation was performed (presumably using the GLMM?)

Done. See lines 153-154.

– It isn't clear why reaction time was not also addressed in the analysis of physiological data, as it was in the hemodynamic data. Some of the bigger effects in Figure 3 were for fast vs. slow RTs.

It is already well known that physiological measures of arousal like pupil size and heart rate covary with RT (see lines 327-329 and 364-385), but the detailed modeling work necessary to link each of these signals to arousal is outside the scope of this paper which focuses on the relationship between the TRR and behavioral performance. We are publishing a separate paper that explores this question, for pupil responses, in detail (Burlingham, Mirbagheri, and Heeger 2022).

We have changed Figure 3C-D, moved the original Figure 3C-D to the supplement (Figure S1), and added a disclaimer about the interpretation of that figure in the caption of Figure S1. We chose not to plot the physiological signals split by RTs in the main text to avoid confusion (see De Gee et al. 2014 and Denison et al. 2020 for further details about the appropriate model). Also see lines 618-622.

– Figure 3c-d- it's initially confusing that the color scheme for fast/slow RT is the same as the correct/incorrect trials in 3b

Agreed. We have moved Figure 3C-D to the supplement and updated the labelling / color scheme to make it clearer. See Figure S1.

– Figure 3f-g- why are there no statistics demonstrating the significance of these results?

Thank you for pointing out this out. Added here: lines 177-178 and 191-195.

– Figure 3G legend: "viability" should be "variability."

Fixed.

– Line 167- an additional line providing details on how the circular standard deviation of the Fourier phase serves as a measure of temporal variability would provide clarity.

See lines 186-189 for the addition.

– Line 168- the authors imply that temporal precision was highest when arousal level was higher. Some brief discussion of this point in the Discussion section would be interesting.

We’ve added discussion of this finding on lines 457-460 of the discussion.

– Line 177 (pupil analysis)- although potentially outside the scope of this paper, it might be valuable to also assess microsaccade rate, which has been linked to arousal, in different task/performance conditions

We are publishing a separate paper about task-related pupil responses and micro-saccades (Burlingham, Mirbagheri, and Heeger 2022). We believe analysis of microsaccades and their relation to arousal to be outside of the scope of the current manuscript.

– Line 189- it should be clearer that all the other physiologic metrics besides pupil were measured in the scanner simultaneously during the fMRI experiment

We have clarified this point on line 63.

– Figure 4: For each physiologic metric, the authors are making multiple comparisons across task/performance conditions but don't discuss any statistical correction for multiple comparisons. Do the significant trends in this figure survive that correction?

We have now performed a correction for multiple comparisons and report the results at the end of the supplement. All of our conclusions are unchanged.

– Figure 4: Was each metric (ex. heart rate, pupil) averaged over the whole trial or some smaller bin within the trial? This detail should be clearer in the methods.

The whole trial. We clarify this on lines 602-603.

If the whole trial was used, could some of the pupil effect be related to the pupillary light reflex evoked by the stimulus flash?

The stimulus was a grating that was equiluminant with the grey background, so it doesn’t evoke a pupillary light reflex. See line 528.

Alternatively, if a smaller bin near the feedback tone was used, this could provide a better estimate of the surprise-related arousal effect the authors mention.

The stimulus and feedback were close together in time (mean RT, 715 ms), so given the amount of time needed to reliably estimate the physiological metrics, this is not possible.

– Line 290 -> pupil dilation has been found to scale with many neuromodulatory (and other cortical and subcortical) systems (e.g. de Gee et al. 2017, Cazettes et al. 2021, Joshi); I find it too strong to equate pupil fluctuations with the LC-NE system.

Agreed. We have modified the discussion on lines 358-362.

– Global signal- if some aspect of the task-related response was present across cortex (consistent with a global arousal signature), then it might be present in the global signal, which was regressed out in this work because it also contains elements of physiologic processes. Potentially outside the scope of this work, but does this trial-to-trial analysis method find a similar TRR in the global signal?

No. We ran the same GLMM (with difficulty, accuracy, fMRI_TO, fMRI_BP, fMRI_TOT, their interactions, and the physiological signals as regressors), and ran the same coefficient tests (looking for modulations with accuracy and difficulty X accuracy). They were both non-significant (p > 0.05). This is now reported in the supplement under the section “Global signal doesn’t show same dependence on task difficulty and behavioral performance that the TRR does.”

– Prior work on TRRs demonstrated "ramp-like" increases and decreases in hemodynamics over minutes between blocks of high and low reward conditions (Cardoso et al. 2019, PLOS). It would be interesting, although potentially outside the scope of this work, if the authors saw similar trends between easy and difficult task blocks or could model this effect in their GLMM by adding trial number within the block as a variable.

We also think this is an interesting and relevant finding of that study of TRRs in macaques. We’ve now discussed it here: lines 467-474.

Unfortunately, unlike the optical imaging signal, the BOLD signal doesn’t have an interpretable baseline, i.e., DC component, so this isn’t advisable. There are a number of low frequency artifacts in the BOLD signal that prevent us from doing such analysis on the timescale of minutes.

– As the authors know, BOLD fMRI studies show robust attention effects in V1 that are much less evident in single-unit studies. This raises the point of how the effects of attention on individual neurons, and in brain areas, relate to the pooled fMRI signal. A similar point could be made here. Perhaps it's not that the effects are smaller across the hierarchy, but rather that the way they manifest across the hierarchy makes them less visible to the MRI signal? Certainly at least one neuromodulator that is thought to relate to arousal, norepinephrine, seems to be more strongly distributed to PFC than to sensory cortical areas (Foote and Morrison, Annual Review of Neurosci 1987, among many suitable papers to reference). I would suggest some discussion of this literature, and idea, as a caveat to this conclusion if it remains.

Thank you for suggesting this idea. We agree that this is a possibility and have discussed this idea and cited this literature on lines 416-422.